# DISENTANGLED ROBOT LEARNING VIA SEPARATE FORWARD AND INVERSE DYNAMICS PRETRAINING

**Wenyao Zhang**[13*]   **Bozhou Zhang**[24*]   **Zekun Qi**[5]   **Wenjun Zeng**[3]   **Xin Jin**[3‡]   **Li Zhang**[24‡]

[1]Shanghai Jiao Tong University   [2]Fudan University   [3]Eastern Institute of Technology, Ningbo
[4]Shanghai Innovation Institute   [5]Tsinghua University

 GitHub Code         HuggingFace

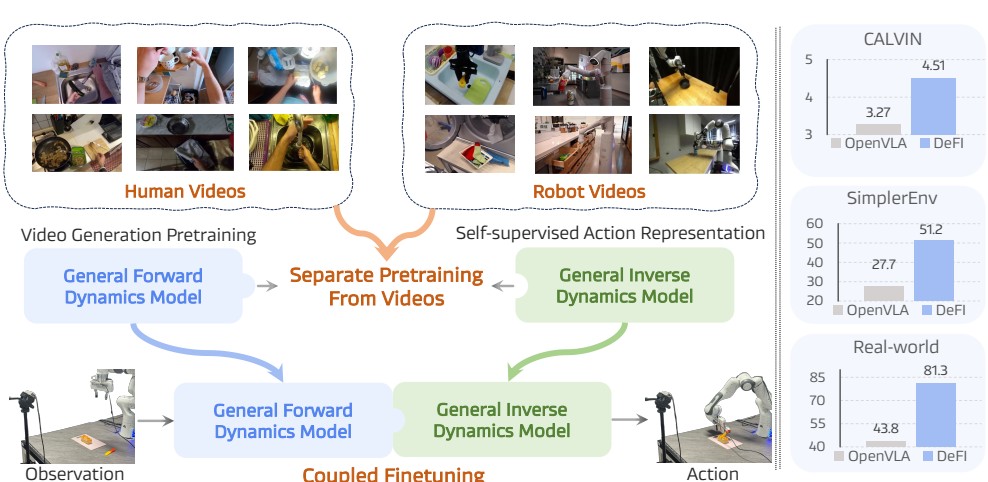

Figure 1: We disentangle robot learning via two decoupled components: a visual forward dynamics model pretrained on large-scale mixed videos via video generation, and an inverse dynamics model pretrained on mixed videos through self-supervised action representation. Then they are coupled during fine-tuning in an end-to-end manner to adapt to downstream tasks. This decoupled pretraining paradigm unleashes the potential of massive action-free videos for policy learning, while retaining robot-specific action grounding, leading to improved success rates across diverse benchmarks.

## ABSTRACT

Vision-language-action (VLA) models have shown great potential in building generalist robots, but still face a dilemma–misalignment of 2D image forecasting and 3D action prediction. Besides, such a vision-action entangled training manner limits model learning from large-scale, action-free web video data. To address these issues, we propose **DeFI**, a novel framework that **De**couples visual **F**orward and **I**nverse dynamics pretraining to exploit respective data sources, wherein video generation and action prediction are disentangled. We introduce the General Forward Dynamics Model (GFDM), pretrained on diverse human and robot videos for future prediction, and the General Inverse Dynamics Model (GIDM), trained via self-supervised learning to infer latent actions from unlabeled video transitions. These models are then integrated into a unified architecture for end-to-end fine-tuning on downstream tasks. In this manner, GFDM and GIDM first shine separately and then cooperate for mutual benefit. Extensive experiments on CALVIN ABC-D and SimplerEnv demonstrate state-of-the-art performance, with DeFI achieving an average task length of 4.51 for CALVIN, 51.2% success rate on SimplerEnv-Fractal benchmark and 81.3% success rate in real-world deployment, significantly outperforming prior methods.

---

*Equal contribution, ‡Corresponding author.

# 1 INTRODUCTION

Vision-language-action (VLA) models (Zitkovich et al., 2023; Kim et al., 2024; Black et al., 2024) have emerged as a promising framework for generalist robots, leveraging the strong visual and language understanding of VLMs (Karamcheti et al., 2024; Beyer et al., 2024) to generate actions with the supervision of massive action-labeled data. A promising line of work (Tian et al., 2024; Zhao et al., 2025; Zhang et al., 2025c) seeks to integrate visual forecasting with action reasoning into an end-to-end architecture, implicitly learning a coupled representation of forward and inverse dynamics, and presents a more impressive success than conventional VLA. However, this paradigm faces two inherent challenges: (i) the competing objectives of 2D video forecasting and 3D action prediction yield unstable training (Tian et al., 2024); (ii) more critically, they hinder the model from fully exploiting these massive action-free human/web videos. We argue that human videos are indispensable for scaling VLA: they are orders of magnitude larger and more diverse than robot demonstrations, and inherently contain rich motion priors across embodiments and tasks. Unlocking their potential is therefore crucial for building truly generalist and scalable robotic agents.

Alternatively, another strategy attempts to bypass this problem by employing a video prediction model pretrained on human and robot videos for forward dynamics learning (Black et al., 2023; Du et al., 2024; Bu et al., 2024a; Liang et al., 2024; Hu et al., 2024; Feng et al., 2025), followed by a simple model for inverse action inference. This strategy reduces dependence on costly action-labeled data and inherits priors from large video generators trained on large-scale corpora. Yet it often overlooks a critical point: *accurate action inference is as important as accurate future prediction, which still needs sufficient data for pretraining to unleash its full ability*. For instance, VPP (Hu et al., 2024) omits the inverse dynamics component entirely, while Vidar (Feng et al., 2025) includes one but treats it contemptuously, without a scalable pretraining recipe—the performance gain stems largely from a powerful video generator (Bao et al., 2024) rather than principled action reasoning. As a result, the inverse dynamics module becomes the bottleneck, unable to fully exploit the predictive power of the forward model and ultimately limiting overall policy performance.

Considering all the above factors, we explore designing an approach to achieve a win-win effect w.r.t. 2D video forecasting and 3D action prediction. To this end, we propose **DeFI**, a novel paradigm that **disentangles robot learning** by decoupling forward and inverse dynamics knowledge pretraining to leverage distinct data sources, then integrating them into a unified, end-to-end architecture to adapt to downstream tasks. Conceptually, both the forward and inverse dynamics modules are pretrained on mixed human and robot data, yet they extract complementary knowledge: the forward dynamics model focuses on capturing motion-level regularities from 2D video forecasting, while the inverse dynamics model emphasizes 3D action reasoning grounded in state transitions. This first separation enables each module to specialize while still benefiting from heterogeneous data, and the following integration yields a scalable and generalizable policy framework. As shown in Figure 1, we first pretrain a visual *general forward dynamics model* (GFDM) built on a video generation model using a mixture of human videos and robot demonstrations. By predicting future video clips conditioned on the current observation and instruction, this GFDM could learn implicit forward dynamics. Crucially, we emphasize *inverse dynamics pretraining is as important as forward dynamics learning*. We therefore introduce a *general inverse dynamics model* (GIDM) with a carefully designed self-supervised recipe that scales to action-free human videos. We cast implicit action inference as a self-supervised representation learning problem: a future video reconstruction objective serves as a proxy that compels the model to distill meaningful latent action codes from visual transitions. This formulation unlocks the use of heterogeneous data to learn inverse dynamics at scale, complementing separated forward-dynamics.

During fine-tuning, we couple the pretrained forward and inverse dynamics models into a unified system that supports end-to-end optimization. This design leverages modality-specific strengths while preserving the benefits of end-to-end learning, enabling strong generalization without relying on massive robot-demonstration datasets. Our comprehensive evaluation, spanning CALVIN ABC-D (Mees et al., 2022b) and SimplerEnv-Fractal (Li et al., 2024b) underscores the framework's efficiency, scalability, and generalization, positioning it as a promising pathway toward next-generation generalist robotic policies. Also, multiple ablation studies are conducted to validate that disentangled robot learning via separate forward and inverse dynamics pretraining could fully exploit the prior knowledge of human videos.

In summary, our main contributions are three-fold: **(i)** A decoupled pretraining paradigm that breaks the reliance of conventional end-to-end VLAs on scarce action annotations, enabling us to exploit abundant, easily available unlabeled video data to learn general physical-world dynamics and action representations; **(ii)** We devise a concise architecture that integrates the separately pretrained forward and inverse dynamics models into a single framework. This design fully leverages action-free human video data while enabling end-to-end fine-tuning on downstream tasks with robot action data **(iii)** DeFI sets a new state of the art on the CALVIN ABC-D benchmark (4.51 average task length), outperforming prior methods by up to 4.2%, and boosts SimplerEnv-Fractal benchmark to 51.2% success rate and real-world experiments to 81.3% success rate. Ablation studies confirm each component's contribution. Furthermore, benefited by pretraining, we only need a few task data to achieve efficient downstream generalization.

## 2 RELATED WORKS

### 2.1 VISION-LANGUAGE-ACTION MODELS

With the vigorous development of Large Language Models (Liu et al., 2023; Karamcheti et al., 2024; Beyer et al., 2024) and the emergence of large-scale robot datasets (O'Neill et al., 2023; Ebert et al., 2021; Khazatsky et al., 2024; Deng et al., 2025), VLA has become a trend in robot learning. RT series (Brohan et al., 2023; Zitkovich et al., 2023; Belkhale et al., 2024) is the pioneering attempt to fine-tune the MLLM on robot demonstration datasets, resulting in strong accuracy and generalization. Based on this, many studies concentrate on improving the accuracy (Kim et al., 2024; Black et al., 2024; Qu et al., 2025; Liang et al., 2025; Xue et al., 2025) and extend to navigation tasks (Zhang et al., 2024b;a). Additionally, many researchers propose to employ multiple knowledge predictions as a multimodal Chain of Thought (COT) to advance the action reasoning ability of VLA. Concretely, prior efforts take several forms. One line of work first plans high-level subtasks and then outputs low-level actions (Belkhale et al., 2024; Lin et al., 2025). Another uses subgoal images or short visual rollouts that anticipate how the scene should evolve (Tian et al., 2024; Zhao et al., 2025; Cen et al., 2025; Wang et al., 2025). A third condition policies on object-centric signals (e.g., bounding boxes) that capture manipulation-relevant dynamics (Deng et al., 2025; Intelligence et al., 2025a). Others learn latent future embeddings or actions that compactly encode forthcoming motor intentions (Bu et al., 2025a; Lyu et al., 2025; Zhang et al., 2025c; Wu et al., 2026). Despite these advances, a central dilemma remains: the misalignment between future-knowledge forecasting and 3D action prediction. Moreover, entangling vision and action during training hampers scaling to action-free web videos. In contrast, DeFI unlocks the potential of large-scale, action-free videos by decoupling the forward and inverse dynamics pretraining, then cooperating in an end-to-end manner for mutual benefit.

### 2.2 ROBOT LEARNING FROM VIDEOS

Research that leverages videos for robot learning typically falls into four branches. First, methods that learn from *explicit human hand/motion labels* (e.g., hand pose, keypoints, contact/trajectory annotations) and transfer these priors to manipulation (Bi et al., 2025; Luo et al., 2025; Kareer et al., 2024; Sun et al., 2026); such labels provide clean supervision but are expensive to scale and brittle under embodiment or camera shifts. Second, methods that *pretrain the policy on mixed videos* and then fine-tune on downstream tasks (Li et al., 2025; Luo & Lu, 2025; Wu et al., 2024; Cui et al., 2024; Majumdar et al., 2023). These methods solely explore using the implicit forward dynamics knowledge in videos to initialize the weights of VLA. Third, methods that *extract latent actions from human videos* to pretrain large VLA models (Ye et al., 2024; Yang et al., 2025; Bjorck et al., 2025; Chen et al., 2025; 2024b), converting video dynamics into compact tokens to amortize over human-scale data. However, this route is indirect—the latents must be consumed by sizable policy models that are costly to pretrain and fine-tune, and the learned codes are not guaranteed to align with the action manifold needed for execution across embodiments. Fourth, *video-as-policy* approaches pretrain a video or latent feature generator on mixed data to imagine future observations and then train a lightweight controller to track those futures (Wen et al., 2024; Hu et al., 2024; Bharadhwaj et al., 2024; Feng et al., 2025; Collins et al., 2025; Black et al., 2023; Du et al., 2024; Zhang et al., 2025a; Tan et al., 2025; Xie et al., 2025; Wen et al., 2023); while these methods exploit abundant action-free footage to learn forward dynamics, a prediction-to-control gap remains. In contrast, our proposed paradigm treats the inverse dynamics model as equally important, and similarly leverages large-scale action-free video data to train it, thereby completing the transfer from prediction to control.

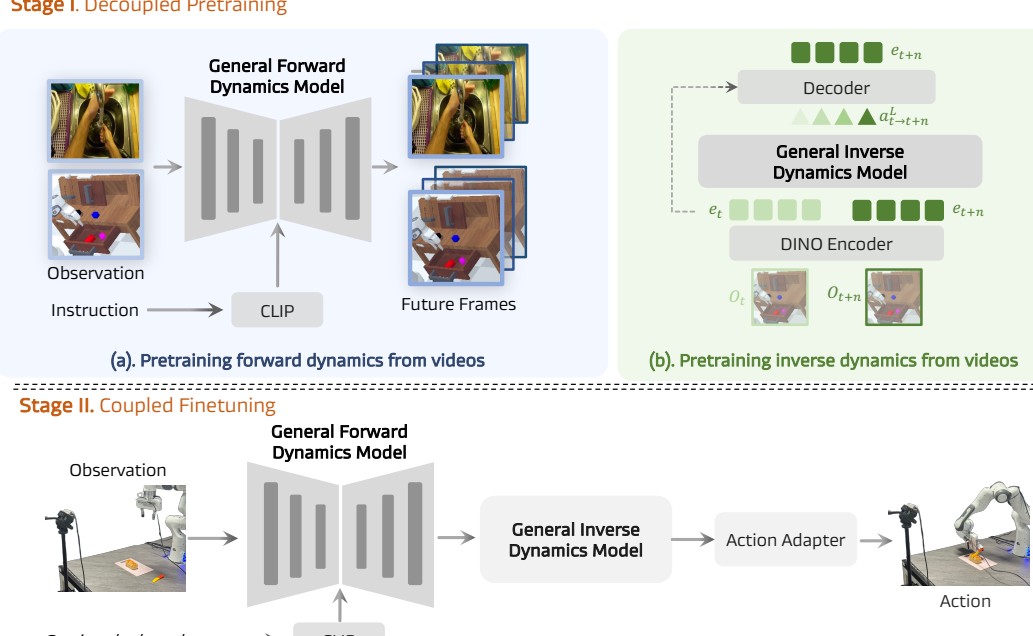

Figure 2: Overall framework of DeFI. **Stage I (Decoupled pretraining):** (a) A general visual forward dynamics model is pretrained with human and robot videos via a video generation objective, predicting future frames from current observations and instructions. (b) In parallel, a general inverse dynamics model is pretrained in a self-supervised manner to map pairs of observations $(o_t, o_{t+n})$ into latent actions, capturing inverse dynamics knowledge without explicit action labels. **Stage II (Coupled finetuning):** The forward and inverse models are coupled, and a diffusion-based adapter is used to generate executable robot action sequences. This two-stage framework unleashes the rich priors of human videos while grounding them in robot data for scalable policy learning.

## 3 METHODOLOGY

As shown in Figure 2, our core idea is to decouple policy learning into two independent knowledge modules: a visual general forward dynamics model that predicts instruction-conditioned future visual states from the current state and instruction in Section 3.1, and a general inverse dynamics component that infers the latent actions responsible for observed visual changes in Section 3.2. Each module is pretrained on large, heterogeneous datasets to absorb complementary priors. We then post-train them together to form a complete policy that maps instructions directly to actions, while supporting end-to-end joint fine-tuning with a small amount of robot data in Section 3.3.

### 3.1 PRETRAIN GFDM TO LEARN FORWARD DYNAMICS

Given a current observation image $o_t$ and a task instruction $l$, the objective of the visual forward dynamics model $\mathcal{F}_\theta$ is to synthesize a short-horizon video $\hat{o}_{t:t+H}$ of length $H+1$. We adopt the stable video diffusion (SVD) model with a CLIP text encoder (Radford et al., 2021) and pretrain it on mixed datasets. The model is composed of three components: (i) a video VAE $(\mathcal{E}, \mathcal{D})$ (2D or 3D) that defines the latent space, (ii) a denoiser $\epsilon_\theta$ (U-Net/Transformer with temporal attention) trained under a latent-diffusion objective. We denote the diffusion time steps by $s \in \{1, \dots, S\}$, distinct from the prediction horizon $H$. With a variance schedule $\{\beta_s\}_{s=1}^S$, define $\alpha_s = 1 - \beta_s$ and $\bar{\alpha}_s = \prod_{i=1}^s \alpha_i$. The forward (add noise) process over the latent video sequence is as follows:

$$q\left(z_{t:t+H}^{(s)} \mid z_{t:t+H}^{(0)}\right) = \mathcal{N}\left(\sqrt{\bar{\alpha}_s}\, z_{t:t+H}^{(0)},\ (1-\bar{\alpha}_s)\mathbf{I}\right), \quad \epsilon \sim \mathcal{N}(0, \mathbf{I}), \tag{1}$$

where the $\epsilon$ denotes the Gaussian noise. The conditioning context is formed from the current observation and instruction:

$$c_t = \big(z_t,\, f_{\text{text}}(l)\big), \quad z_t = \mathcal{E}(o_t), \tag{2}$$

where $z_t$ is obtained by encoding the current image. The denoiser is optimized via noise prediction (optionally with $v$-parameterization):

$$\mathcal{L}_{\text{diff}}(\theta) = \mathbb{E}_{z_{t:t+H}^{(0)},\, s,\, \epsilon} \left\| \epsilon - \epsilon_\theta\big(z_{t:t+H}^{(s)},\, s,\, c_t\big) \right\|_2^2. \tag{3}$$

During the inference stage, starting from Gaussian noise, a sampler (e.g., DDIM or DPM-Solver) generates the latent forecast:

$$\hat{z}_{t:t+H} = \mathcal{F}_\theta(z_t, f_{\text{text}}(l)).$$

Nevertheless, fully denoising an entire explicit video remains computationally expensive, as most of the cost is wasted on reconstructing pixel-level details irrelevant to manipulation. In contrast, the key signal for control lies in motion dynamics rather than appearance. Recent research (Zhu et al., 2025; Hu et al., 2024) further suggests that a generative model's features after a single denoising step already contain sufficient motion information to guide downstream action planning. Inspired by this, we freeze the pretrained GFDM and restrict the denoising process to a single step, yielding efficient predictions of future latent embeddings. Notably, for a robot with multiple camera views, such as a third-view and a wrist camera, we predict the future videos for each view independently.

## 3.2 Pretrain GIDM to Learn Inverse dynamics

To learn the knowledge of inverse dynamics from mixed videos in a fully unsupervised manner, we develop a proxy task to pretrain the general inverse dynamics model $\mathcal{I}_\theta$. Specifically, we start with a pair of consecutive video frames $o_t, o_{t+n}$, separated by a frame interval n, then extract a pair of latent states $e_t$ and $e_{t+n}$ using the DINOv2 (Oquab et al., 2024) visual encoder. We ensure a uniform time interval of approximately 1 second across diverse datasets. The general inverse dynamics model consists of an encoder built upon a spatial-temporal Transformer (Xu et al., 2020) with causal temporal masks, and a VQ-VAE codebook that enables vector-quantized action representation. We concatenate a set of learnable action queries $q_a \in \mathbb{R}^{N \times d}$ with predefined dimension $d$, along the sequence dimension with the DINO embeddings of the current and future frames as well as the instruction embeddings extracted by T5 (Raffel et al., 2020), and feed them into the GIDM:

$$\tilde{a}_{t \to t+n}^L = \mathcal{I}_\theta\big(e_t, e_{t+n}, l, q_a\big),$$

Following LAPA (Ye et al., 2024), we train the model using the VQ-VAE objective (Van Den Oord et al., 2017), which implicitly quantizes the latent actions. The nearest quantized representation is retrieved from a discrete embedding codebook:

$$\hat{a}_{t \to t+n}^L = \mathcal{VQ}_\theta\big(\tilde{a}_{t \to t+n}^L\big).$$

This formulation allows the latent action to be represented as discrete tokens from a vocabulary space $|C|$, making it straightforward for vision-language models to predict actions. The quantized latent action is then passed to a decoder composed of Spatial Transformers, which predicts the DINO features of the future frame. $\hat{e}_{t+n}$ The training objective minimizes the mean-squared error (MSE) loss between the predicted future states $\hat{e}_{t+n}$ and ground-truth states $e_{t+n}$.

## 3.3 Finetuning the Coupled GFDM and GIDM in an End-to-end Manner

During finetuning, we keep the GFDM $\mathcal{F}_\theta$ frozen because it was pretrained on large-scale data that already covers the downstream domain; further finetuning on the much smaller downstream split would erode these dynamics priors and hurt generalization. The frozen GFDM then serves as a stable backbone that provides temporally consistent future video representations encoding long-horizon dynamics. A lightweight MLP then projects them onto the input manifold of the GIDM, ensuring representational compatibility between forward prediction and inverse reasoning. The GIDM $\mathcal{I}_\phi$ is then optimized to interpret these aligned latents and infer latent actions that capture the underlying motion. To extract richer spatiotemporal representations, we further use a video former (Hu et al., 2024) to obtain intermediate-layer features from the GFDM. Finally, we fuse the MLP-projected features with the video-former features and feed the combined representation into the diffusion-based action adapter, initialized from a 30M DiT-B, is trained to translate latent actions into executable robot commands. This finetuning stage therefore couples the three modules and allows their objectives—future prediction, action inversion, and low-level control—to be jointly aligned.

### 3.4 INFERENCE PROCESS

At inference time, the GFDM receives the current observation $o_t$ and language instruction $l$, generates a sequence of predicted future video features $\hat{z}_{t:t+H}$ through a single-step denoising process. These features capture the anticipated future scene evolution and establish a dynamics-aware context for downstream action reasoning. The MLP then projects these future embeddings into the input space of the GIDM, which combines them with the current latent state to infer the latent action sequence The diffusion-based action adapter then conditions on these latent actions and produces the final executable control commands.

## 4 EXPERIMENTS

In this section, we conduct extensive experiments on both simulated and real-world environments to evaluate the effectiveness of DeFI as shown in Figure 3.

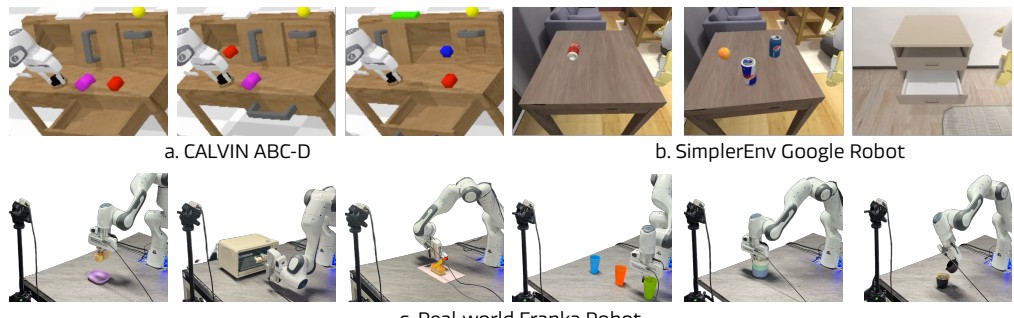

Figure 3: Experiments setup on CALVIN ABC-D, SimplerEnv Google Robot and real-world Franka Robot. We evaluate DeFI across 3 simulation environments.

### 4.1 IMPLEMENTATION DETAILS

In the pretraining stage, we first train the general forward dynamics model on a diverse collection of datasets spanning both human videos (Goyal et al., 2017; Grauman et al., 2022) and robotic manipulation data (Mees et al., 2022b; O'Neill et al., 2023). In parallel, the general inverse dynamics model is pretrained on large-scale human egocentric datasets, including Ego4D (Grauman et al., 2022) and Open X-Embodiments (O'Neill et al., 2023). During fine-tuning, we freeze the pretrained forward dynamics model to preserve its generalization capability and generate 16-frame future predictions. All experiments are conducted on NVIDIA H100 GPUs. Detailed implementation and training protocols are provided in Appendix A.2.

### 4.2 MANIPULATION BENCHMARKS ON CALVIN

**Experiment setup and baseline.** CALVIN (Mees et al., 2022b) is a simulated benchmark designed for learning long-horizon, language-conditioned robot manipulation policies. It comprises four distinct manipulation environments and provides over six hours of teleoperated play data per environment, captured from multiple sensors including static and gripper-mounted RGB-D cameras, tactile images, and proprioceptive readings. We focus on the challenging ABC-D setting, where the model is trained in the ABC environment and evaluated in the unseen D environment, then report the success rate of every track and the average length of 5 tasks. We compare our model with the latest state-of-the-art generalist manipulation policies, including OpenVLA (Kim et al., 2024), Robovlm (Li et al., 2024a), $\pi_0$ (Black et al., 2024), GR1 (Wu et al., 2024), UP-VLA (Zhang et al., 2025b), Seer (Tian et al., 2024), SuSIE (Black et al., 2023), CLOVER (Bu et al., 2024b) and VPP (Hu et al., 2024). For fairness, we evaluate our approach in two setups: a static (third) view and a multi-view setting that combines static and wrist cameras.

**Quantitative results and analysis.** As shown in Table 1, DeFI can be effectively adapted to tasks in the CALVIN ABC-D environments under different view settings. Our method surpasses OpenVLA, $\pi_0$, and GR1, which directly project the RGB images into action signals, revealing that leveraging a

Table 1: **CALVIN ABC-D results.** We present the average success computed over 1000 rollouts for each task and the average number of completed tasks to solve 5 instructions consecutively (Avg. Len.). DeFI shows significant superiority over baselines. The best results are **bolded**. *We reproduced results of $\pi_{0.5}$, GR00T N1 and OpenVLA-OFT on CALVIN.

| View | Method | 1 | 2 | 3 | 4 | 5 | Avg. Len. ↑ |
|---|---|---|---|---|---|---|---|
| Third View | SuSIE (Black et al., 2023) | 87.0 | 69.0 | 49.0 | 38.0 | 26.0 | 2.69 |
| | CLOVER (Bu et al., 2024b) | 96.0 | 83.5 | 70.8 | 57.5 | 45.4 | 3.53 |
| | UniVLA (Bu et al., 2025b) | **95.5** | 85.8 | 75.4 | 66.9 | 56.5 | 3.80 |
| | **DeFI** | 92.9 | **87.2** | **81.2** | **75.0** | **68.4** | **4.05** |
| Multi-View | GR-1 (Wu et al., 2024) | 85.4 | 71.2 | 59.6 | 49.7 | 40.1 | 3.06 |
| | OpenVLA (Kim et al., 2024) | 91.3 | 77.8 | 62.0 | 52.1 | 43.5 | 3.27 |
| | Vidman (Wen et al., 2024) | 91.5 | 76.4 | 68.2 | 59.2 | 46.7 | 3.42 |
| | $\pi_0*$ (Black et al., 2024) | 93.8 | 85.0 | 76.7 | 68.1 | 59.9 | 3.84 |
| | $\pi_0.5*$ (Intelligence et al., 2025b) | 94.8 | 87.4 | 78.2 | 71.7 | 64.3 | 3.97 |
| | GR00T N1* (Bjorck et al., 2025) | 94.2 | 86.1 | 79.6 | 73.9 | 66.8 | 4.01 |
| | UP-VLA (Zhang et al., 2025b) | 92.8 | 86.5 | 81.5 | 76.9 | 69.9 | 4.08 |
| | Seer (Tian et al., 2024) | 96.3 | 91.6 | 86.1 | 80.3 | 74.0 | 4.28 |
| | VPP (Hu et al., 2024) | 96.5 | 90.9 | 86.6 | 82.0 | 76.9 | 4.33 |
| | **DeFI** | **97.9** | **94.2** | **90.7** | **87.0** | **81.2** | **4.51** |

powerful forward dynamics model to predict future actions would benefit current action reasoning. We compare with UniVLA, which extracts latent action labels from human videos, and then pretrains the VLA model on the large-scale datasets. It demonstrates that decoupling forward and inverse dynamics model pretraining is more effective than solely extracting latent action as pseudo labels to pretrain VLA. Additionally, DeFI surpasses UP-VLA and Seer, which integrate the visual/latent feature forecasting and action reasoning into a single VLA framework. The results demonstrate that disentangling the forward and inverse dynamics models and pretraining them on mixed videos separately would fully exploit the power of action-free videos and benefit the robot action reasoning. Compared to methods that use video generation models' predicted videos as input, like SuSIE, CLOVER and VPP, our model significantly achieves more accurate control, demonstrating that *accurate action inference is as important as accurate future prediction and a powerful inverse dynamics model leads to better performance.* Furthermore, we can find that our method is more effective in long-horizon tasks than previous methods, because our visual forward dynamics model can predict future videos and leverage a powerful GFDM to resolve the actions.

**Data efficiency.** Collecting robot data is both time-consuming and labor-intensive, making data efficiency crucial for robot learning. We evaluate our method on the CALVIN ABC-D benchmark, using 10%, 20%, 50%, and 100% of the available data to fine-tune pretrained policies. The results, shown in Figure 4, demonstrate that our method consistently enhances policy performance across varying data scales. Notably, under data-scarce conditions with only 10% of the training data, the pretrained policy achieves an 18% relative improvement in average task length on CALVIN ABC-D compared to VPP (Hu et al., 2024). Moreover, our method requires only about 60% of the data on CALVIN ABC-D to surpass the previous state-of-the-art baseline. These results highlight the potential of DeFI in scenarios with limited finetuning data and further push the upper bound of robot learning by introducing massive low-cost human videos.

## 4.3 MANIPULATION BENCHMARKS ON SIMPLERENV-FRACTAL

**Experiment setup and baseline.** SimplerEnv (Li et al., 2024b) features WidowX and Google Robot setups, providing diverse manipulation scenarios with varied lighting, colors, textures, and robot camera pose conditions, thereby bridging the visual appearance gap between real and simulated environments. We compare our model on the Fractal branch (Google Robot) with the latest state-of-the-art generalist manipulation policies, including Octo (Team et al., 2024), TraceVLA (Zheng et al., 2024), and OpenVLA (Kim et al., 2024).

**Quantitative results and analysis.** Table 2 presents the SimplerEnv experimental results on the Fractal branch. DeFI also achieves state-of-the-art performance on Google robot multitasks, with an average success rate of 51.2% and 45.4% on visual matching and variant aggregation settings, respectively. However, DeFI underperforms on certain tasks. We attribute this to domain shift: the

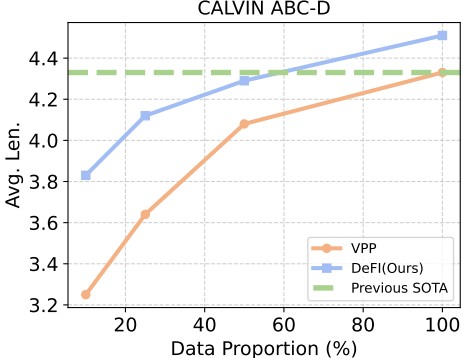

Figure 4: **Data efficiency** of DeFI's performance on CALVIN using different proportions of the action-labeled downstream data.

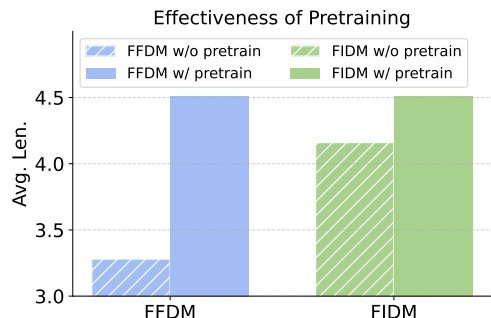

Figure 5: Ablation study for the effectiveness of decoupled forward and inverse pretraining.

Table 2: **Evaluation results across different policies on SimplerEnv**. We evaluate DeFI on 3 tasks on the Google Robot in SimplerEnv.

| | SimplerEnv on Google Robot Tasks | | | | | | | |
|---|---|---|---|---|---|---|---|---|
| Model | Visual Matching | | | | Variant Aggregation | | | |
| | Pick Coke Can | Move Near | Open/Close Drawer | Avg. | Pick Coke Can | Move Near | Open/Close Drawer | Avg. |
| Octo-Base (Team et al., 2024) | 17.0% | 4.2% | 22.7% | 16.8% | 0.6% | 3.1% | 1.1% | 1.1% |
| TraceVLA (Zheng et al., 2024) | 28.0% | 53.7% | **57.0%** | 42.0% | **60.0%** | 56.4% | **31.0%** | 45.0% |
| OpenVLA (Kim et al., 2024) | 16.3% | 46.2% | 35.6% | 27.7% | 54.5% | 47.7% | 17.7% | 39.8% |
| **DeFI** | **54.2%** | **60.7%** | 38.6% | **51.2%** | 53.9% | **58.2%** | 24.0% | **45.4%** |

visual GFDM is pretrained on real-world datasets (Fractal (Brohan et al., 2023)) and kept frozen during finetuning, which restricts it to predicting real-world images. This mismatch propagates to the inverse dynamics model, causing it to generate erroneous actions.

## 4.4 REAL-WORLD EXPERIMENTS

**Experiment setup and baselines.** As shown in Figure 6, we use the Franka Panda arm to conduct experiments evaluating the effectiveness of our method in the real world. In our setup, two RealSense D415 cameras capture RGB images: one provides a third-person view, and the other is mounted on the gripper. We collected 1,600 trajectories for 8 tasks, as shown in Table 3. In the experimental setup, each trial allows a maximum of 20 consecutive attempts. All objects are randomly positioned on the table surface. A trial is considered successful if the robotic arm grasps the target object within the specified attempts; in placement tasks, success further requires transferring the object onto a designated plate. For fair comparison, we finetune Diffusion Policy (Chi et al., 2023), Octo-Base (Team et al., 2024), OpenVLA (Kim et al., 2024) and DeFI on collected demonstration datasets.

**Quantitative results and analysis.** As presented in Table 3, DeFI outperforms previous methods. Specifically, in simple single-task scenarios (place, open, and close), all the policies exhibit good performance($> 50\%$). However, in moderately

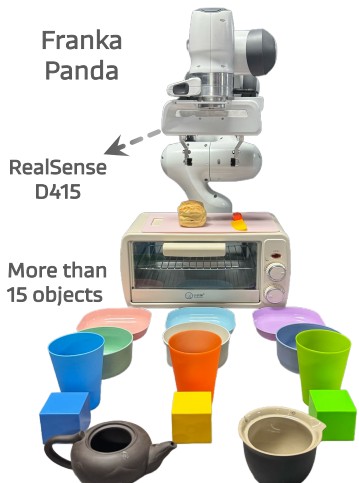

Figure 6: Real-world robot setup.

complex tasks (cut & stack), where the models need to make the robot take or stack different colors and sizes of objects, most policies, such as DP, Octo, and OpenVLA, struggle with manipulation, frequently encountering issues like object and order misidentification. Our method surpasses these approaches thanks to the powerful prediction and generalization capabilities of the pretrained visual general forward dynamics model. Furthermore, in the complex long-horizon and accurate control task (pour water), our method demonstrates strong performance, accurately executing tasks like grasping up a teapot and pouring water into a cup, which relies on the powerful pretrained inverse dynamics

Table 3: **Real-world evaluation** with the Franka Robot across eight tasks.

| Method | Success Rate (%) | | | | | | | |
|---|---|---|---|---|---|---|---|---|
| | Place | Open | Close | Cut | Stack Bowl | Stack Cube | Stack Bottle | Pour Water | Average |
| Diffusion Policy (Chi et al., 2023) | 70.0 | 40.0 | 70.0 | 50.0 | 45.0 | 35.0 | 40.0 | 35.0 | 48.2 |
| Octo-Base (Team et al., 2024) | 55.0 | 35.0 | 60.0 | 20.0 | 30.0 | 25.0 | 30.0 | 20.0 | 34.4 |
| OpenVLA (Kim et al., 2024) | 50.0 | 40.0 | 65.0 | 40.0 | 30.0 | 35.0 | 45.0 | 45.0 | 43.8 |
| **DeFI** | **90.0** | **75.0** | **100.0** | **80.0** | **80.0** | **70.0** | **80.0** | **75.0** | **81.3** |

Table 4: **Performance comparison** with or without decoupled pretraining.

| Addition Type | Task completed in a row | | | | | |
|---|---|---|---|---|---|---|
| | 1 | 2 | 3 | 4 | 5 | Len. |
| GFDM w/o pre | 88.0 | 77.6 | 62.4 | 56.8 | 43.2 | 3.28 |
| GIDM w/o pre | 96.0 | 88.8 | 83.2 | 76.8 | 71.2 | 4.16 |
| All w/ pre | **97.9** | **94.2** | **90.7** | **87.0** | **81.2** | **4.51** |

Table 5: **Performance comparison** with or without human videos.

| Addition Type | Task completed in a row | | | | | |
|---|---|---|---|---|---|---|
| | 1 | 2 | 3 | 4 | 5 | Len. |
| All w/o h.v. | 93.6 | 91.2 | 88.0 | 82.4 | 79.2 | 3.92 |
| GFDM w/o h.v. | 96.0 | 91.2 | 85.6 | 77.6 | 68.8 | 4.19 |
| GIDM w/o h.v. | 93.6 | 91.2 | 88.0 | 82.4 | 79.2 | 4.34 |
| All w/ h.v. | **97.9** | **94.2** | **90.7** | **87.0** | **81.2** | **4.51** |

model. Overall, DeFI achieves a higher average success rate, showcasing robust real-world operation capabilities.

## 4.5 ABLATION STUDY

In this section, we investigate the following questions under the multi-view setting to thoroughly evaluate the ability of our model:

**Q1: What is the impact of the decoupled pretraining stage?** As shown in Table 4 and Figure 5, the GFDM without pretraining achieves an average task length of 3.28, benefiting from the prior knowledge of the video generation model. However, it still suffers from limited prediction quality on robot videos. Without pretraining, the GIDM achieves an average length of 4.16, while pretraining the inverse dynamics branch provides stronger action guidance. Incorporating the full decoupled pretraining further improves performance to 4.51. These results highlight the importance of large-scale decoupled pretraining on robot and human data for stable optimization and better generalization.

**Q2: What impact does human video have?** As shown in Table 5, the GFDM without human videos achieves an average task length of 4.19. When the GIDM is used without human videos, the performance improves slightly to 4.34. Incorporating human videos during pretraining further increases the average task length to 4.51, yielding relative gains of +0.17 over GIDM and +0.32 over GFDM. These results indicate that the massive scale and diverse human video data provide valuable motion priors that complement robot demonstrations and enhance generalization.

**Q3: How does the quality and format of the predicted image affect performance?** As shown in Table 6, we study the trade-off between generation quality and latency by varying the number of denoising steps. While additional denoising slightly improves visual fidelity, it increases the latency. In our DeFI, a single denoising step requires approximately 150 ms per inference, whereas five denoising steps take around 250 ms, resulting in significantly slower performance. Notably, a single denoising step already captures sufficient semantic information about future frames, and further steps do not yield improvements in manipulation performance. We also test replacing the SVD-based video backbone with a DINO-based generative model. While DINO features converge faster and better match the GIDM feature space, they cannot integrate well with existing video-generation frameworks or leverage their pretrained knowledge, leading to inferior performance than the SVD baseline. It's a promising way to explore stronger DINO (Zhou et al., 2024) or other latent embedding prediction models (Bardes et al., 2023; Assran et al., 2025) to better understand the upper bound.

**Q4: What impact do architectural variants of the inverse dynamics model have?** As shown in Table 7, to rigorously evaluate the impact of inverse dynamics model architecture on overall policy performance, we compare GIDM against several common architectural variants: (i).a simple multilayer perceptron (MLP) that directly maps concatenated current and future state embeddings to actions, (ii).a Transformer that directly maps concatenated current and future state embeddings to actions, and (iii).our GIDM, which discretizes the continuous action space via a causal transformer with vector quantization using VQ-VAE. Our GIDM outperforms all alternatives. The MLP baseline

Table 6: **Performance comparison** of different settings. "5 Steps" indicates five denoising steps in GFDM, while "DINO" denotes a DINO-based generative model used as GFDM.

| Method | Task completed in a row | | | | | |
|---|---|---|---|---|---|---|
| | 1 | 2 | 3 | 4 | 5 | Len. |
| 5 Steps | **98.4** | **95.2** | 89.6 | 82.4 | 79.2 | 4.45 |
| DINO | 91.2 | 81.6 | 74.4 | 65.6 | 57.6 | 3.70 |
| Ours | 97.9 | 94.2 | **90.7** | **87.0** | **81.2** | **4.51** |

Table 7: **Performance comparison** of different inverse dynamics model architectures.

| Method | Task completed in a row | | | | | |
|---|---|---|---|---|---|---|
| | 1 | 2 | 3 | 4 | 5 | Len. |
| MLP | 89.6 | 80.8 | 66.4 | 56.0 | 48.8 | 3.42 |
| Transformer | 97.6 | 90.4 | 83.2 | 77.6 | 73.6 | 4.22 |
| Ours | **97.9** | **94.2** | **90.7** | **87.0** | **81.2** | **4.51** |

Table 8: **Performance comparison** of different discretization methods for stabilizing inverse dynamics learning.

| Method | Task completed in a row | | | | | |
|---|---|---|---|---|---|---|
| | 1 | 2 | 3 | 4 | 5 | Len. |
| G.M. | 94.1 | 90.3 | 84.7 | 78.5 | 72.9 | 4.12 |
| S.B. | 93.2 | 89.1 | 82.3 | 75.4 | 69.8 | 3.98 |
| C.L.A | 94.5 | 91.0 | 86.2 | 80.1 | 74.7 | 4.20 |
| Ours | **97.9** | **94.2** | **90.7** | **87.0** | **81.2** | **4.51** |

Table 9: Ablation study on finetuning GFDM, GIDM, and action adapter.

| Method | Task completed in a row | | | | | |
|---|---|---|---|---|---|---|
| | 1 | 2 | 3 | 4 | 5 | Len. |
| Adapter Only | 95.5 | 92.0 | 87.2 | 81.1 | 75.7 | 4.33 |
| GFDM+Adapter | 95.6 | 92.4 | 88.1 | 82.5 | 76.2 | 4.35 |
| GIDM+Adapter | 97.9 | 94.2 | 90.7 | 87.0 | 81.2 | 4.51 |
| All Train | 96.8 | 93.1 | 88.4 | 83.2 | 78.0 | 4.40 |

fails to capture complex actions, while the Transformer accumulates errors. These results demonstrate that the design of the inverse dynamics model critically affects performance, and that our discrete action space approach with self-supervised training effectively learns a structured latent action space for precise action generation.

**Q5: How do different discretization strategies affect performance?** To evaluate the effectiveness of different discretization methods, we conduct an ablation study comparing four strategies: Gaussian Mixture(G.M), Simple Binning(S.B), Continuous Latent Action(C.L.A), and our proposed Discrete VQ-VAE method. The VQ-VAE in our method serves not only as a discretization tool but also as an information-bottleneck mechanism, which stabilizes the learning of inverse dynamics. This quantization step helps prevent future-state leakage into the decoder, ensuring the model learns meaningful action representations instead of relying on low-level visual shortcuts. The results of our ablation study, shown in Table 8, demonstrate that the discrete VQ-VAE method outperforms the other discretization strategies, providing significant improvements in task performance.

**Q6: What happens when different modules (FDM/IDM/Adapter) are partially fine-tuned?** As shown in Table 9, Adapter Only already achieves strong performance despite having very few trainable parameters, indicating that the pretrained GFDM provides a strong and expressive latent space. FDM+Adapter yields only minor improvements, suggesting that forward prediction alone is insufficient for reliable action inference. Although "All Train" updates FDM, IDM, and the action adapter jointly, its performance is lower than IDM+Adapter(Ours) because joint optimization introduces representation instability and gradient interference. When GFDM is finetuned, its latent outputs change throughout training, causing the input distribution of IDM to drift. As a result, the IDM must continually adapt to shifting representations, making it much harder to learn a stable and accurate action-recovery function. Additionally, the frozen forward dynamics model could provide better generalization for action reasoning.

## 5 CONCLUSION

We presented DeFI, a framework that decouples visual forward and inverse dynamics pretraining to reconcile the misalignment between 2D video forecasting and 3D action prediction while enabling learning from large-scale, action-free web videos. DeFI comprises a General Forward Dynamics Model for future prediction from diverse human and robot videos and a General Inverse Dynamics Model that infers latent actions from unlabeled video transitions. The two models are integrated into a unified architecture and fine-tuned end-to-end on downstream tasks, allowing them to first specialize independently and then cooperate for mutual benefit. This method consistently enhances both simulated and real tasks, showing that decoupling forward and inverse dynamics offers a scalable and effective path for VLA systems trained on Internet-scale video.

## ETHIC STATEMENT

This work complies with the ICLR Code of Ethics. It does not involve human or animal subjects, nor the use of private or sensitive data. All datasets are publicly available and used under their respective licenses. The research raises no direct ethical or legal concerns, and the authors are committed to responsible and fair use of the proposed methods.

## REPRODUCIBILITY STATEMENT

We have made every effort to ensure the reproducibility of our work. The proposed model, implementation details, and evaluation protocols are described in detail in the main paper and appendix. All datasets used are publicly available and properly referenced. To further support reproducibility, we will release the source code.

## ACKNOWLEDGEMENTS

This work was supported in part by New Generation Artificial Intelligence-National Science and Technology Major Project (2025ZD0123004), Ningbo grant (2025Z038) and National Natural Science Foundation of China (Grant No. 62376060), Grants of NSFC 62302246, ZJNSFC LQ23F010008, Ningbo 2023Z237 & 2024Z284 & 2024Z289 & 2023CX050011 & 2025Z038 & 2025Z059, and supported by High Performance Computing Center at Eastern Instituteof Technology and Ningbo Institute of Digital Twin.

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

# A APPENDIX

## A.1 THE USE OF LARGE LANGUAGE MODELS

Large language models are used solely as writing assistants for grammar refinement and expression polishing. They do not contribute to research ideation, methodology design, experiments, or analysis.

## A.2 IMPLEMENTATION DETAILS

**GFDM Pretraining Details.** For pretraining the general forward dynamics model, we use a mixture of robot video datasets (Open X-Embodiment (O'Neill et al., 2023), CALVIN (Mees et al., 2022b)) and human video datasets (Something-Something-v2 (Goyal et al., 2017), Ego4D (Grauman et al., 2022)). To appropriately balance the contributions of different datasets, we adopt varying sampling ratios. Detailed dataset information is provided in Table 11.

**GIDM Pretraining Details.** For pretraining the general inverse dynamics model, we use a subset of the Open X-Embodiment dataset (O'Neill et al., 2023) containing single-arm end-effector control. Although actions and proprioceptive states are available in these robot datasets, we exclude them during pretraining and rely only on episode frames and text instructions. We further incorporate open-world human videos, specifically egocentric recordings of daily activities from the Ego4D dataset (Grauman et al., 2022). Except for the SimplerEnv benchmark (Li et al., 2024b), which replicates the environment of the Fractal dataset (Brohan et al., 2023), none of the downstream evaluation environments (e.g., CALVIN (Mees et al., 2022b)) are seen during pretraining, thereby requiring strong generalization capabilities from the model. The dataset composition and sampling ratios are detailed in Table 12.

**Coupled Finetuning Details** For the coupled finetuning stage, we freeze the general forward dynamics model while finetuning the general inverse dynamics model and the latent action adapter. Training is conducted on the CALVIN-ABC dataset for evaluation on the CALVIN benchmark (Mees et al., 2022b), and on the Fractal dataset (Brohan et al., 2023) for evaluation on the SimplerEnv benchmark (Li et al., 2024b).

We summarize the training and model parameters of each component of DeFI in Table 10.

## A.3 MODEL ARCHITECTURE

**General forward dynamics model.** We adopt the open-sourced Stable Video Diffusion (SVD) (Blattmann et al., 2023) as the general forward dynamics model. We further enhance it by incorporating language instructions through CLIP (Radford et al., 2021) and adjusting the output video resolution to $256 \times 256$, aligning with the resolution of robot datasets (O'Neill et al., 2023; Mees et al., 2022b).

**General inverse dynamics model.** We adopt a Transformer architecture as the general inverse dynamics model and train it in the DINO feature (Oquab et al., 2024) space to obtain semantically rich representations, following prior work (Bu et al., 2025b). The pseudo-code for the pretraining process is shown in Algorithm 1. Following previous works (Ye et al., 2024; Chen et al., 2024b; Bu et al., 2025b), we use a Transformer decoder to reconstruct the features of future frames when training the general inverse dynamics model. However, the Transformer decoder is discarded during the fine-tuning stage of the coupled general forward dynamics model (GFDM) and general inverse dynamics model (GIDM).

**Diffusion-based action adapter.** We adopt a Diffusion Transformer architecture as the action adapter to decode latent action features into robot actions. The language instruction, encoded by the CLIP encoder, is combined with the latent action features obtained from the general inverse dynamics model and serves as conditioning for the action denoising process, which generates the final robot actions.

Table 10: Training and model parameters used in our DeFI.

| Train parameter | Value |
|---|---|
| GPU | NVIDIA H100 |
| Number of GPUs | 8 |
| Pretraining time of GFDM | 3 days |
| Pretraining time of GIDM | 1.5 days |
| Finetuning time on CALVIN | 0.5 days |
| Training memory on CALVIN | 64G |
| Inference memory on CALVIN | 7G |
| Batch size | 32 |
| Learning rate | $1 \times 10^{-4}$ |
| Weight decay | $1 \times 10^{-2}$ |
| Optimizer | AdamW |
| Pretraining epochs | 20 |
| Finetuning epochs | 12 |
| **Model parameter** | **Value** |
| *General Forward Dynamics Model* | |
| Model type | Stable Video Diffusion |
| Image size | $256 \times 256$ |
| Predicted future frames | 16 |
| *General Inverse Dynamics Model* | |
| Model type | Transformer |
| Feature dimension | 768 |
| Vocabulary size of VQ codebook | 128 |
| Number of layers | 16 |
| *Action Adapter* | |
| Model type | Diffusion Transformer |
| Feature dimension | 384 |
| Number of layers | 12 |
| Sampling steps | 10 |
| Action dimension | 7 |

---

**Algorithm 1:** General Inverse Dynamics Model Training

---

**Input:** Current frame $o_t$, future frame $o_{t+n}$, language instruction $L$
**Output:** Predicted DINO feature of $o_{t+n}$
$F_t \leftarrow$ DINOEncoder$(o_t)$ ;                   // DINO feature of current frame
$F_{t+n} \leftarrow$ DINOEncoder$(o_{t+n})$ ;            // DINO feature of future frame
$F_L \leftarrow$ TextEncoder$(L)$ ;                      // Instruction embedding
$H \leftarrow$ Spatial-temporal Transformer$(F_t, F_{t+n}, F_L)$ ;  // General Inverse Dynamics
 Model
$A \leftarrow$ VQ-VAE$(H)$ ;                             // Latent action feature
$\hat{F}_{t+n} \leftarrow$ TransformerDecoder$(F_t, A)$ ;        // Decoded future DINO feature
$\mathcal{L}_{\text{pred}} \leftarrow$ Loss$(\hat{F}_{t+n}, F_{t+n})$ ;            // Prediction loss
$\mathcal{L}_{\text{VQ}} \leftarrow$ Loss$(H, A)$ ;                          // VQ-VAE loss
$\mathcal{L} \leftarrow \mathcal{L}_{\text{pred}} + \mathcal{L}_{\text{VQ}}$ ;                          // Total loss

---

## A.4 EXPERIMENTS

### A.4.1 REAL-WORLD EXPERIMENTS.

As shown in Figure 17, success is recorded only if both the grasping and placement operations are completed within the allowed attempts. For the articulated object manipulation tasks(open & close), the microwave is randomly placed in front of the robotic arm. The experiment is considered

Table 11: The general forward dynamics model of our DeFI is trained on a mixture of data from the Open X-Embodiment (O'Neill et al., 2023) and CALVIN (Mees et al., 2022b) robot video datasets, as well as the Ego4D (Grauman et al., 2022) and Something-Something-v2 (Goyal et al., 2017) human video datasets. The proportions are normalized to sum to 100%.

| Category | Training dataset mixture | Proportion |
|---|---|---|
| **Robot Videos** | Fractal (Brohan et al., 2023) | 30% |
| | Bridge (Ebert et al., 2021; Walke et al., 2023) | 10% |
| | CALVIN-ABC (Mees et al., 2022b) | 30% |
| **Human Videos** | Something-Something-v2 Goyal et al. (2017) | 15% |
| | Ego4D Grauman et al. (2022) | 15% |

Table 12: The general inverse dynamics model of our DeFI is trained on a mixture of data from the Open X-Embodiment (O'Neill et al., 2023) robot video dataset and the Ego4D (Grauman et al., 2022) human video dataset. The proportions are normalized to sum to 100%.

| Category | Training dataset mixture | Proportion |
|---|---|---|
| **Robot Video** | Fractal (Brohan et al., 2023) | 16.3% |
| | Kuka (Kalashnikov et al., 2018) | 7.4% |
| | Bridge (Ebert et al., 2021; Walke et al., 2023) | 8.0% |
| | Taco Play (Mees et al., 2022a) | 4.1% |
| | Jaco Play (Dass et al., 2023) | 0.7% |
| | Berkeley Cable Routing (Luo et al., 2024) | 0.4% |
| | Roboturk (Mandlekar et al., 2018) | 3.3% |
| | Viola (Zhu et al., 2023b) | 1.3% |
| | Berkeley Autolab UR5 (Chen et al., 2024a) | 1.6% |
| | Toto (Zhou et al., 2023) | 2.8% |
| | Language Table (Lynch et al., 2023) | 6.1% |
| | Stanford Hydra Dataset (Belkhale et al., 2023) | 6.2% |
| | Austin Buds Dataset (Zhu et al., 2022) | 0.4% |
| | NYU Franka Play Dataset (Cui et al., 2022) | 1.2% |
| | Furniture Bench Dataset (Heo et al., 2023) | 3.4% |
| | UCSD Kitchen Dataset (Darkhalil et al., 2022) | 0.1% |
| | Austin Sailor Dataset (Nasiriany et al., 2022) | 3.0% |
| | Austin Sirius Dataset (Liu et al., 2022) | 2.3% |
| | DLR EDAN Shared Control (Quere et al., 2020) | 0.1% |
| | IAMLab CMU Pickup Insert (Saxena et al., 2023) | 1.3% |
| | UTAustin Mutex (Shah et al., 2023) | 3.0% |
| | Berkeley Fanuc Manipulation (Zhu et al., 2023a) | 1.1% |
| | CMU Stretch (Mendonca et al., 2023) | 0.2% |
| | BC-Z (Jang et al., 2022) | 10.3% |
| | FMB Dataset (Lynch et al., 2023) | 9.8% |
| | Dobbe (Shafiullah et al., 2023) | 2.0% |
| **Human Video** | Ego4D (Grauman et al., 2022) | 3.5% |

successful if the door displacement exceeds 5 cm, indicating effective interaction. For the cutting task, the robot has to take the knife to cut the bread.

### A.4.2 ADDITIONAL ABLATION STUDY

**Q7: What is the scaling behavior of human data?** To evaluate the scaling behavior of human video data, we conduct an ablation study, as shown in Table 7 and Figure 8. The results show that performance improves as the amount of human video increases, with marginal gains becoming smaller at larger scales but no saturation is observed, suggesting the potential for further improvements with even larger datasets.

Figure 7: **Performance** with increasing amounts of human video data for dynamics pretraining.

Figure 8: Scaling results with different amounts of human video used for dynamics pretraining.

| Method | Task completed in a row | | | | | |
|---|---|---|---|---|---|---|
| | 1 | 2 | 3 | 4 | 5 | Len. |
| 0.0 | 92.4 | 85.6 | 78.0 | 70.2 | 63.1 | 3.92 |
| 0.2 | 94.6 | 88.2 | 83.3 | 77.5 | 71.9 | 4.16 |
| 0.4 | 96.4 | 91.0 | 86.0 | 80.7 | 75.4 | 4.30 |
| 0.6 | 96.1 | 92.4 | 87.4 | 82.6 | 77.3 | 4.36 |
| 0.8 | 97.5 | 93.3 | 89.2 | 84.1 | 78.9 | 4.43 |
| **1.0** | **97.9** | **94.2** | **90.7** | **87.0** | **81.2** | **4.51** |

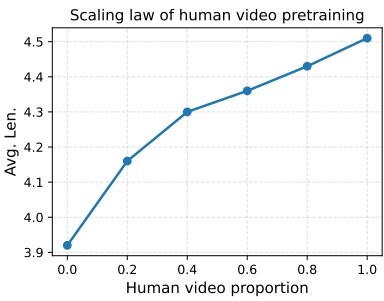

**Q8: The visualization of different denoising step.** As shown in Figure 9, we present the visualization of the forward dynamics model across different denoising steps(1, 10, 20). Although a single denoising step may blur the background regions, the motion-critical information—such as the object trajectory and the robot end-effector path—remains well preserved, which explains why task performance remains stable. In addition, finetuning the inverse dynamics model and the adapter further mitigates potential information loss, enabling the model to operate reliably even under this aggressive optimization.

**Q9: The statistics of failure cases.** We examined 200 failure cases on CALVIN (failures defined as not completing the task within 280 steps). The errors can be grouped into two major categories:

**(i) Forward-dynamics failures (62%):** More challenging scenarios occur in contact-rich or cluttered interactions, where the FDM may generate hallucinated or physically implausible predictions and mismatch videos for multi-view. In these cases, the prediction error becomes too large for the IDM to compensate. Figure 10 shows such a scenario, where accumulated inconsistencies in the imagined future scene ultimately mislead the controller. These failures indicate that long-horizon consistency and contact modeling remain bottlenecks for world-model-based approaches.

**(ii) Inverse-dynamics failures (38%):** Even when the predicted future is accurate, the IDM may still produce incorrect actions—for instance, misplacing objects, failing to grasp, or causing collisions—as shown in Figure 11. These failures highlight that action inference is an equally critical component and can bottleneck overall performance independent of the world model's accuracy.

This highlights our motivation: accurate inverse dynamics is as essential as accurate forward prediction for reliable control.

## A.5    INFERENCE LATENCY

As shown in Table 13, we evaluate the inference time of our model on the CALVIN (Mees et al., 2022b) benchmark. The inference time of each component of the model is reported, averaged over five runs.

Table 13: The inference time of our DeFI is measured on an NVIDIA GeForce RTX 4090 GPU, averaged over five runs.

| Model part | Inference time |
|---|---|
| General Forward Dynamics Model | 86.1ms |
| General Inverse Dynamics Model | 42.9ms |
| Action Adapter | 24.3ms |

## A.6    DISCUSSIONS AND FUTURE WORK

Our model introduces a new framework that disentangles robot learning into a general forward dynamics model and a general inverse dynamics model, enabling full utilization of large-scale action-free videos from both humans and robots. This represents a fundamental improvement over existing vision-language-action (VLA) architectures, particularly in scenarios where embodied intelligence data is costly. For the limitation, our model does not incorporate a large language model, and therefore

lacks the ability to support language-based interaction. Interaction and embodied reasoning (Qi et al., 2025; 2024) are crucial for complex robotic tasks. For future work, we aim to integrate our GFDM and GIDM with a large language model as a foundation understanding module, enabling the model to unify prediction, interaction, and action execution capabilities.

## A.7 QUALITATIVE RESULTS

**Heatmap of GIDM.** As shown in Figure 12, we visualize the attention maps of the general inverse dynamics model (GIDM) to demonstrate its ability to capture actions from both robot and human videos. The results indicate that the model consistently attends to the robot arm or human arm across different time steps, enabling it to extract latent actions that guide the generation of executable actions. This highlights the benefit of large-scale pretraining in grounding the model's action understanding.

**Qualitative results on the benchmark.** As shown in Figure 13 and Figure 14, we visualize the results of the CALVIN benchmark on long-horizon tasks. Our DeFI performs well on sequences consisting of five consecutive tasks. Similarly, Figure 15 and Figure 16 present results on the SimplerEnv benchmark for the tasks "pick coke can" and "close drawer", where DeFI completes the tasks coherently according to the given instructions.

**Qualitative results on the real-world environments.** As shown in Figure 17, we visualize the results of the real-world environments across eight tasks.

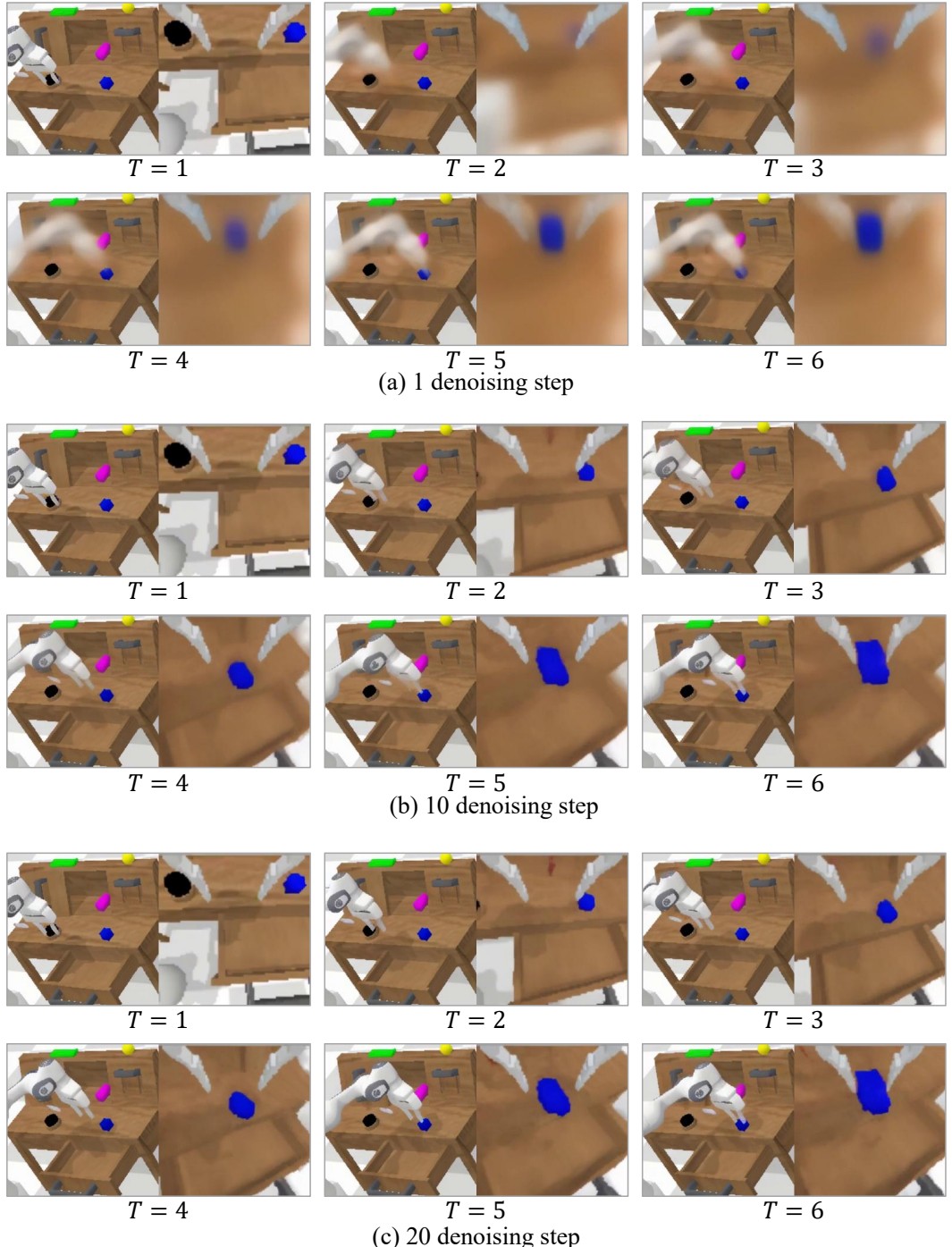

Figure 9: Qualitative results of different denosing step for the forward dynamics model.

Go push the pink block right

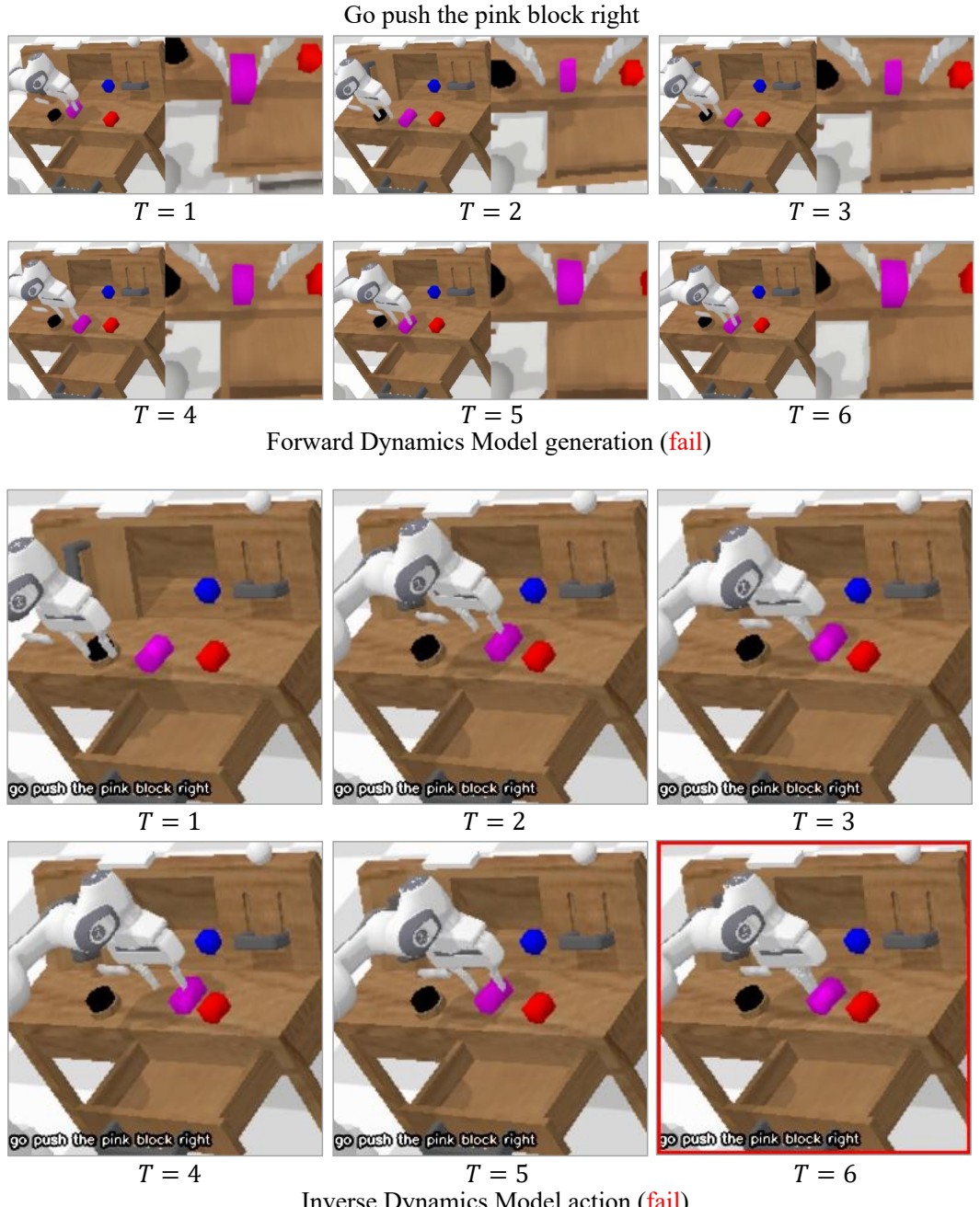

Figure 10: Qualitative results of failure cases.

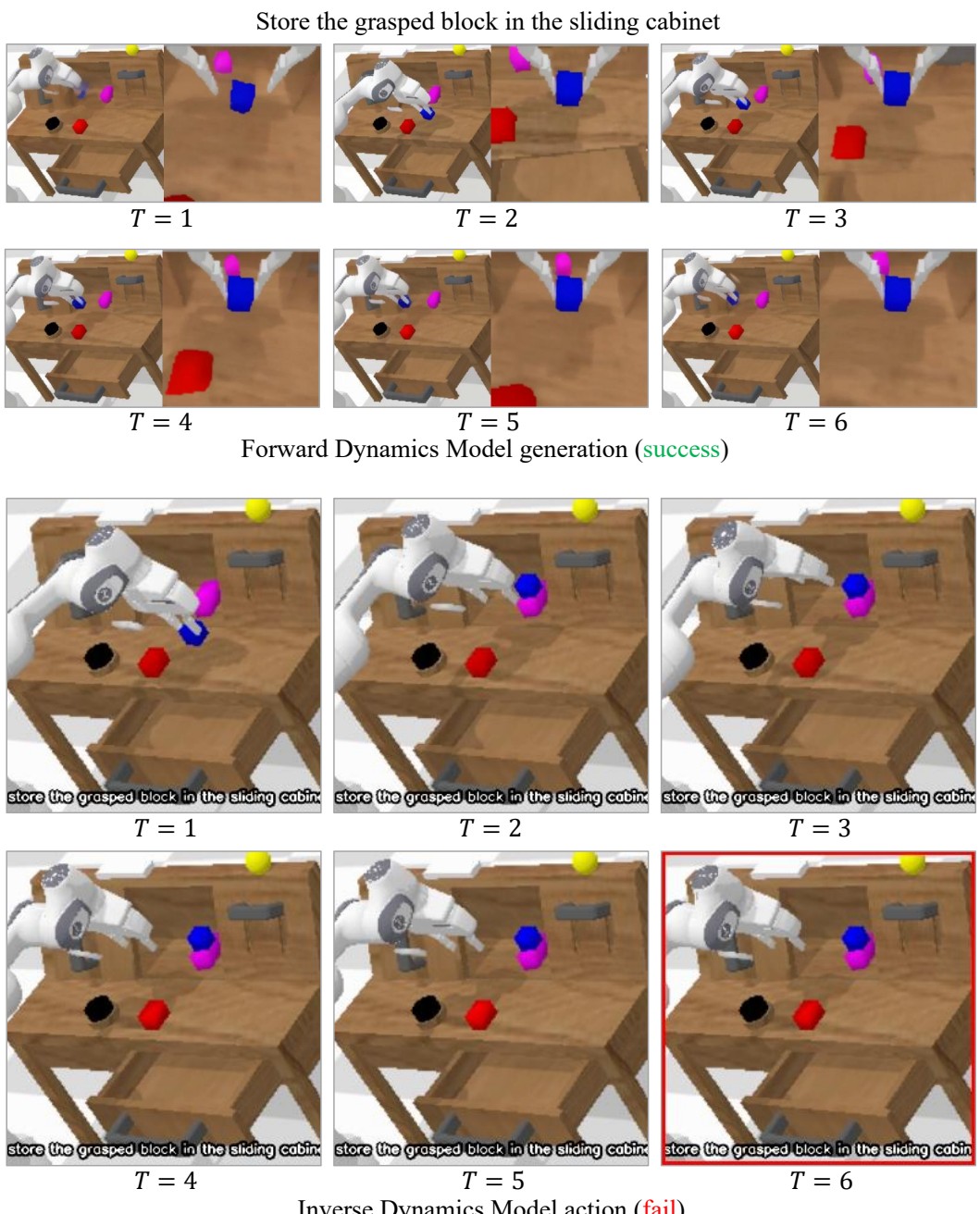

Figure 11: Qualitative results of failure cases.

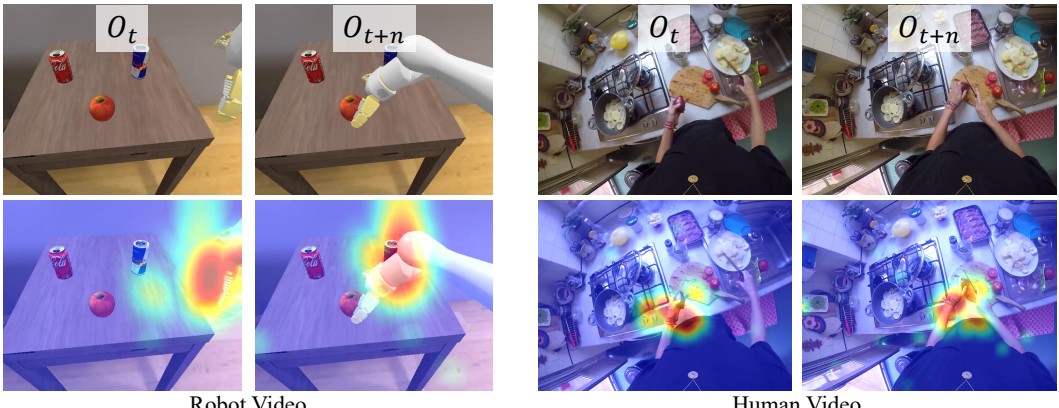

Figure 12: Qualitative attention heatmap results of the general inverse dynamics model on robot and human videos.

go push the red block left

grasp and lift the red block

store the grasped block in the sliding cabinet

push the sliding door to the right side

use the switch to turn off the light bulb

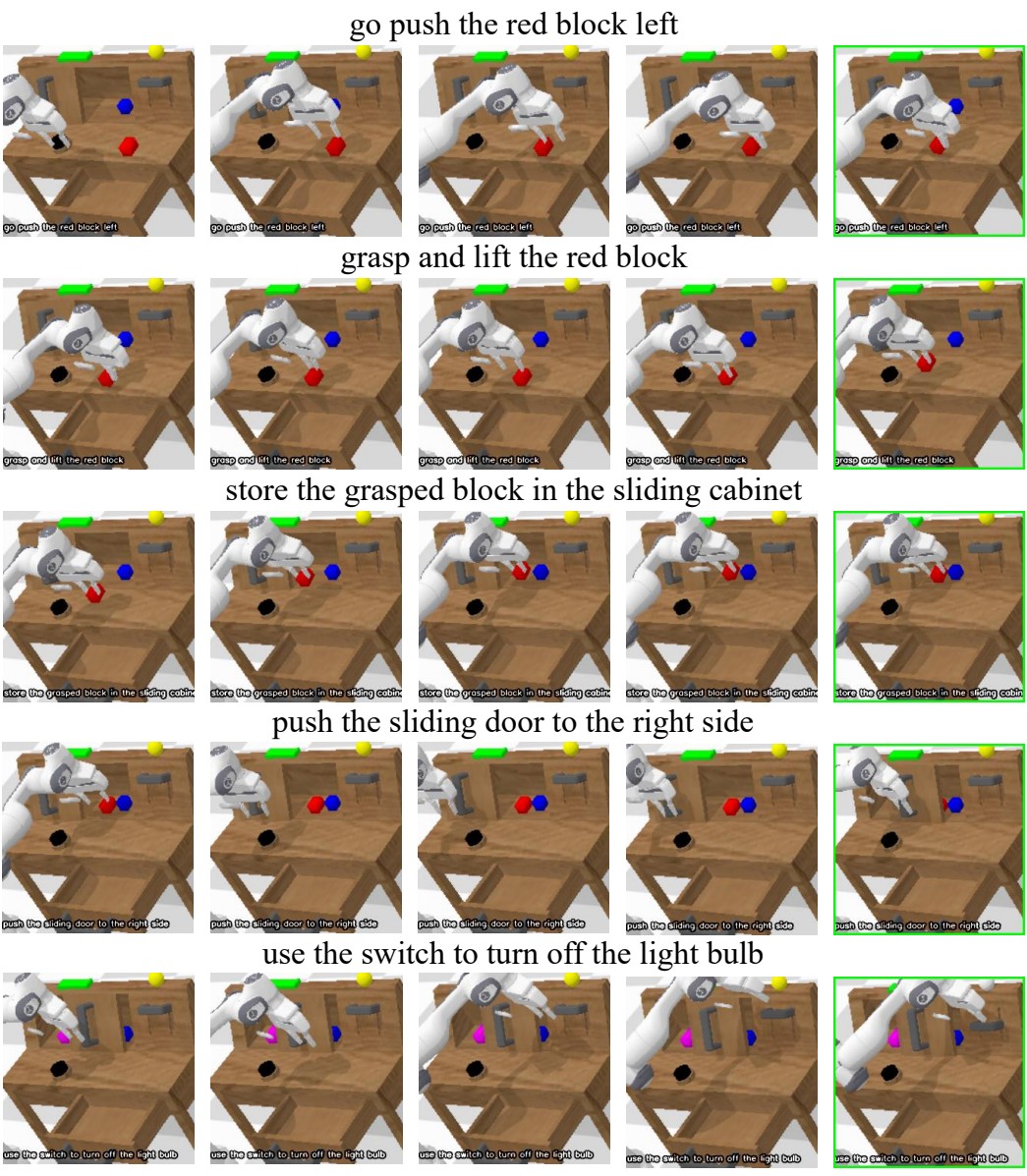

Figure 13: Qualitative results of the CALVIN long-horizon task.

take the red block and rotate it to the right

push the sliding door to the right side

grasp and lift the red block

store the grasped block in the sliding cabinet

press the button to turn on the led light

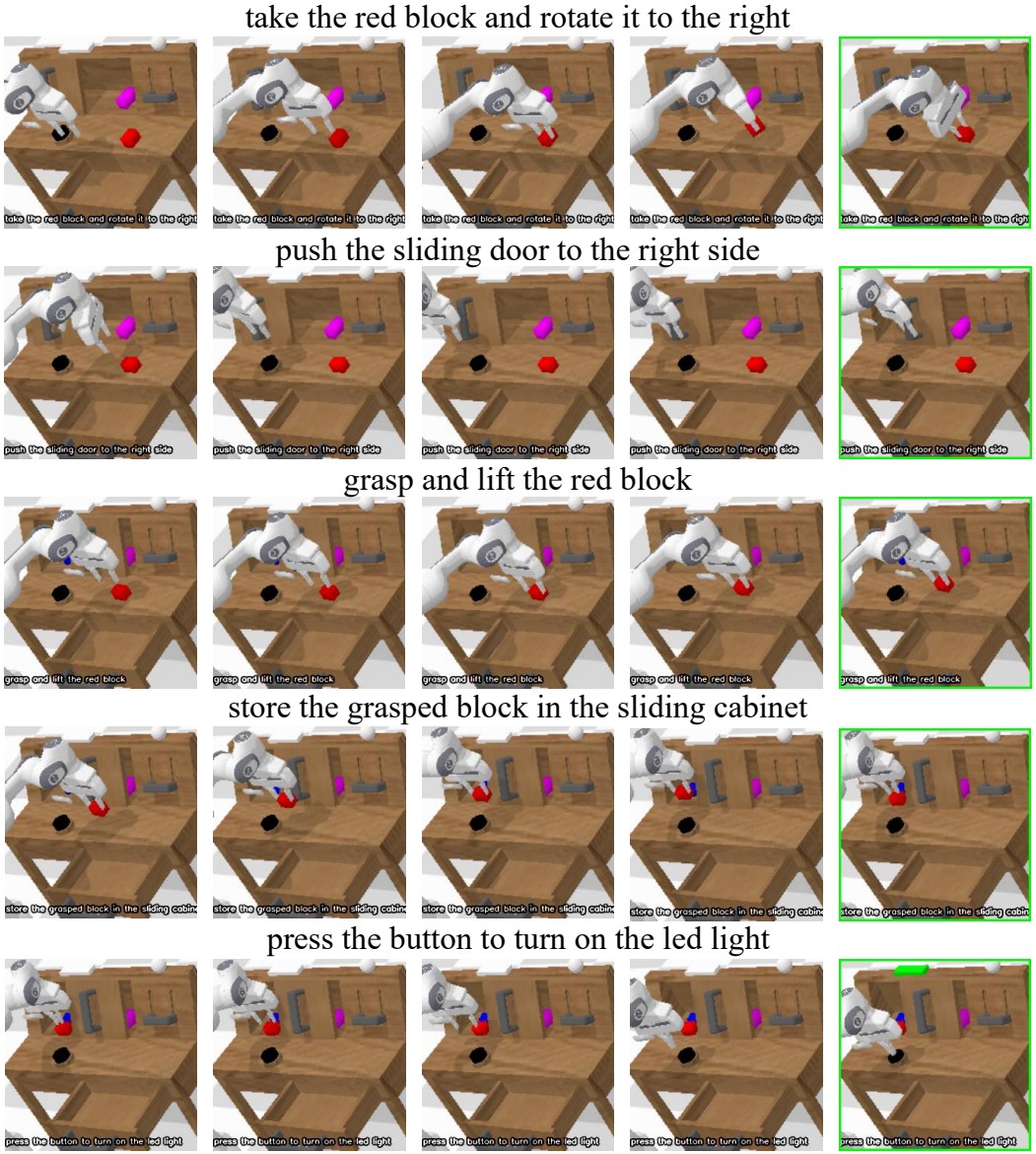

Figure 14: Qualitative results of the CALVIN long-horizon task.

Pick Coke Can

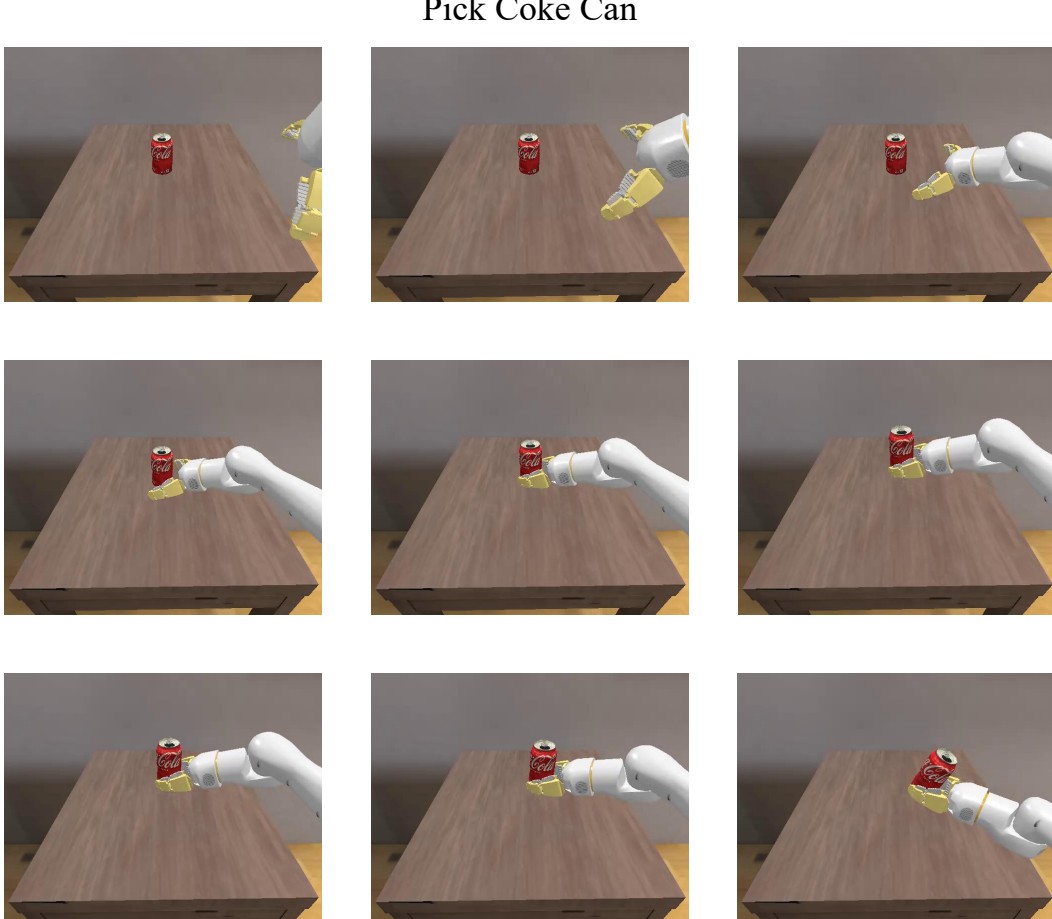

Figure 15: Qualitative results of SimplerEnv evaluation on Google Robot.

## Close Drawer

Figure 16: Qualitative results of SimplerEnv evaluation on Google Robot.

Real-world experiments

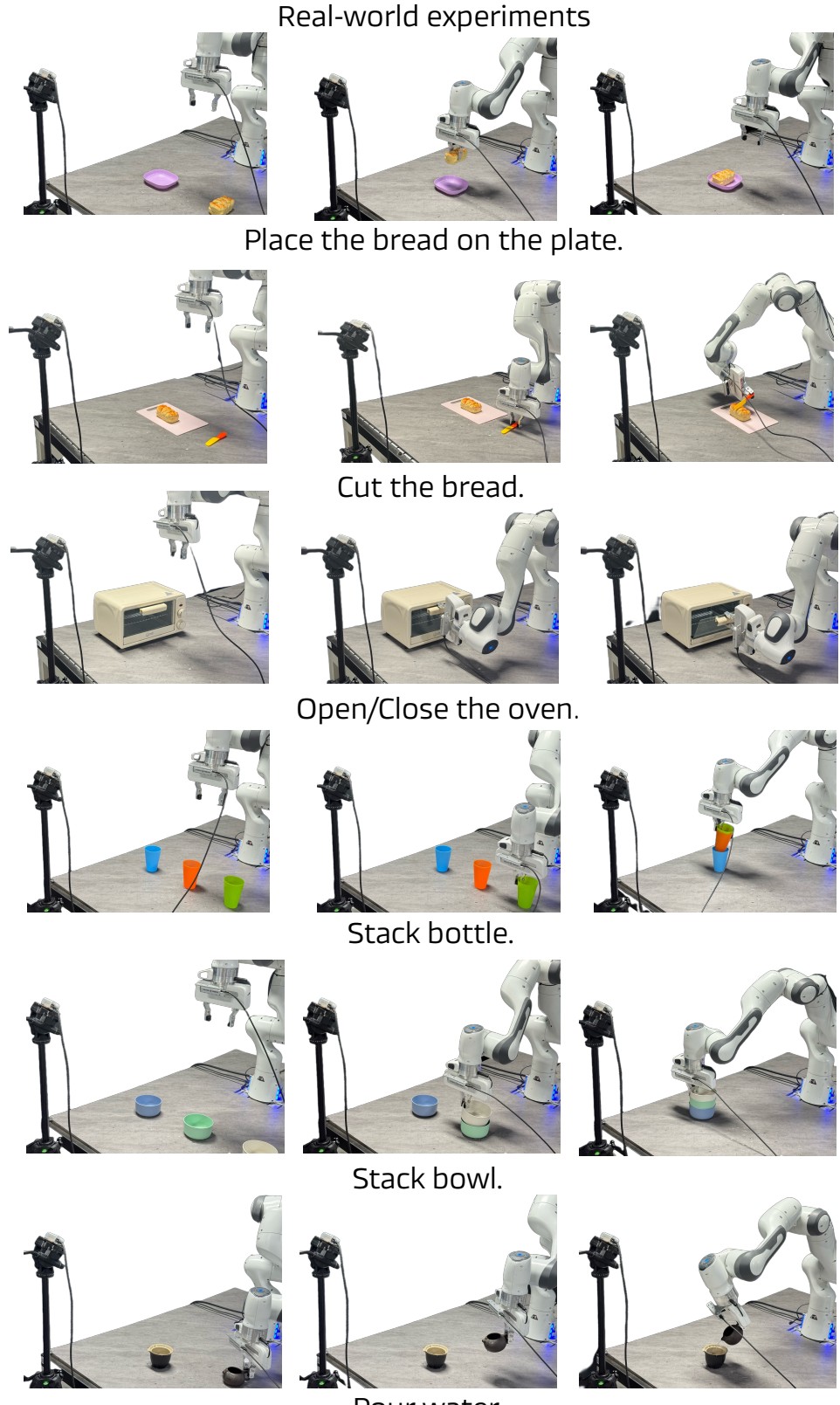

Figure 17: Qualitative results of real-world experiments.

