# OpenReview forum: "Disentangled Robot Learning via Separate Forward and Inverse Dynamics Pretraining"
_ICLR.cc/2026/Conference — ICLR 2026 Poster_

### Official Review · Reviewer_zL2E · 2025-11-01

**Soundness:** 2
**Presentation:** 3
**Contribution:** 3
**Rating:** 4
**Confidence:** 3

**Summary:**

The paper proposes DeFI, a framework that decouples forward and inverse dynamics pretraining for robot policy learning. A Foundation Forward Dynamics Model (FFDM) is pretrained via video diffusion on human + robot videos to model visual dynamics, while a Foundation Inverse Dynamics Model (FIDM) learns latent actions self-supervisedly from video transitions. The two are later coupled and fine-tuned on downstream tasks (CALVIN, SimplerEnv, Franka). DeFI reports higher average task length and success rates than prior VLA baselines.

**Strengths:**

There is clear conceptual novelty in separating forward / inverse pretraining, and synthesis of diffusion video modeling and latent-action quantization.

The experiments are extensive, with solid ablations showing component effects.

The targeted problem is significant, which is scaling robot learning with action-free human videos.

**Weaknesses:**

Unfair empirical comparison (major): DeFI is fine-tuned on target datasets (CALVIN, SimplerEnv, Franka), whereas baselines such as OpenVLA and $\pi_0$ appear evaluated as off-the-shelf checkpoints trained on their original dataset, which are zero-shot evaluated on the author's benchmarks against DeFI's fine-tuned checkpoint.

No zero-shot results: Despite “foundation” framing, all evaluations use fine-tuning ≥10 % of labeled data; zero-shot capability is not demonstrated.

**Questions:**

I highly appreciate the authors for proposing a novel idea in a potentially very impactful area. The paper is well-written and thorough. However, the concerns over the unfair comparison above is a major concern of mine. If the authors can adequately address why the comparison is, or has to be set up this way, that would help me a lot in re-evaluating the decision.

---

> ### Author Response · Authors · 2025-11-21
> **Official Comment by Authors**
>
> We thank the reviewer for the insightful comments and constructive suggestions. Below we address each concern in detail. We hope our clarifications and new experiments resolve your concerns, and we would be very grateful if you could consider raising your score after reading this rebuttal.
>
> ### [**_W1&Q1_: The confusion about unfair empirical comparison.**]
> Our comparison is fair in all experiments, because OpenVLA and π₀ are not evaluated in a zero-shot setting. Instead, both baselines are finetuned on the same downstream datasets as DeFI, including CALVIN, SimplerEnv, and our real-world Franka dataset.
> Although OpenVLA and π₀ are pretrained on large-scale robotics datasets, they still require finetuning on downstream benchmarks because the camera viewpoints, simulation environments, and objects used in our experiments differ from those in their pretraining data. For fairness, we finetune all baselines and DeFI under the same data, training setup and evaluation protocol as previous work π₀[1], SpatialVLA[2], CoT-VLA[3]. Hence, our comparisons are fair and consistent across all methods.
>
> >[1].Black, Kevin, et al. "$\pi_0 $: A Vision-Language-Action Flow Model for General Robot Control." arXiv preprint arXiv:2410.24164 (2024).
>
> >[2].Qu, Delin, et al. "Spatialvla: Exploring spatial representations for visual-language-action model." arXiv preprint arXiv:2501.15830 (2025).
>
> >[3].Zhao, Qingqing, et al. "Cot-vla: Visual chain-of-thought reasoning for vision-language-action models." Proceedings of the Computer Vision and Pattern Recognition Conference. 2025.
>
> ### [**_W2_:  The meaning of "foundation".**]
> The term "foundation" in our paper refers to pretraining on large-scale data, which enables strong transfer learning to downstream tasks. As demonstrated in the manuscript, our method achieves significantly better performance after fine-tuning on tasks like CALVIN, SimplerEnv, and real-world experiments. While other methods like π₀ and OpenVLA have also been pretrained on similar or larger datasets, they still underperform compared to our approach. Thus, our method excels in transfer learning, even if it is not evaluated in a zero-shot setting.

---

> > ### Comment · Reviewer_zL2E · 2025-11-24
> >
> > I thank the authors for confirming that the baselines are set up as finetuned checkpoints.
> >
> > An additional question - Have you tried, or any reason it was hard to try, OpenVLA-OFT as opposed to the original OpenVLA?

---

> ### Author Response · Authors · 2025-11-25
>
> Thanks for your response.
> We will incorporate your feedback accordingly and are currently running the official OpenVLA-OFT experiments on the CALVIN benchmark, with results expected within the next 1–3 days. We will include the full comparison in the updated rebuttal and the revision as soon as the experiments complete, and if the findings further strengthen our method, we would greatly appreciate your reconsideration of our rating.

---

> > ### Comment · Reviewer_zL2E · 2025-11-26
> >
> > Thank you and looking forward to the results. That would be the last piece I need to adjust the score.

---

> > > ### Author Response · Authors · 2025-11-28
> > >
> > > Following the reviewer’s suggestion, we additionally include π0.5, GR00T N1 and OpenVLA-OFT results on CALVIN, shown below (this will be added as Table 1 in the revision):
> > > | Addition Type    | 1    | 2    | 3    | 4    | 5    | Len. |
> > > |------------------|------|------|------|------|------|------|
> > > |π0 |93.8 | 85.0 | 76.7| 68.1 | 59.9| 3.84|
> > > |π0.5     | 94.8 | 87.4 | 78.2 | 71.7 | 64.3 | 3.97 |
> > > | GR00T N1    | 94.2 |86.1| 79.6| 73.9 |66.8 | 4.01 |
> > > | OpenVLA-OFT | 95.7 | 88.2 | 82.4 | 74.5 | 70.7 | 4.12 |
> > > |**DeFI(Ours)**|**97.9** | **94.2** | **90.7** | **87.0**| **81.2** |**4.51**|
> > > **Analysis**: Our method significantly outperforms π0.5, GR00T N1 and OpenVLA-OFT, all of which directly map RGB observations to actions. This comparison highlights a key advantage of DeFI: **pretraining on large-scale mixed data provides strong prior knowledge, and the forward dynamics model predicts future states to supply richer guidance for action inference**, resulting in substantially better performance. This trend is also supported by concurrent works such as CoT-VLA [1], DreamVLA [2], and VPP [3].
> > >
> > > >[1].Zhao, Qingqing, et al. "Cot-vla: Visual chain-of-thought reasoning for vision-language-action models." Proceedings of the Computer Vision and Pattern Recognition Conference. 2025.
> > >
> > > >[2].Zhang, Wenyao, et al. "DreamVLA: A Vision-Language-Action Model Dreamed with Comprehensive World Knowledge." The Thirty-ninth Annual Conference on Neural Information Processing Systems.
> > >
> > > >[3].Hu, Yucheng, et al. "Video Prediction Policy: A Generalist Robot Policy with Predictive Visual Representations." Forty-second International Conference on Machine Learning.

---

### Official Review · Reviewer_Hekh · 2025-11-01

**Soundness:** 3
**Presentation:** 3
**Contribution:** 3
**Rating:** 6
**Confidence:** 4

**Summary:**

The authors address the problem of VLA models having a misalignment of 2d image forecasting and 3d action prediction.  The paper proposes DeFI (Decoupled visual Forward and Inverse dynamics) pretraining to disentangle video generation and action prediction.
Two models are introduced: The Foundation Forward Dynamics model (FFDM) is pretrained on human / robot videos for future predictio; and the Foundation Inverse Dynamics model (FIDM) is trained via self-supervised learning to infer latent actions from unlabeled video transitions.  The FFDM and FIDM are then integrated together in an end-to-end finetuning for downstream tasks. Performance on various manipulation benchmarks is presented.

**Strengths:**

- The disentanglement of forward and inverse dynamics learning enables leveraging distinct data sources.
- Enables pretraining with action-free internet-scale video data. The pretrained FFDM can then be coupled with a FIDM that is trained for actions with different embodiments if necessary.
- For robots with multiple cameras, the FFDM predicts future videos for each view independently.

**Weaknesses:**

- The claim of new state of art on Calvin only outperforms prior methods by 4.2%.  It will be good to justify if and why this is a significant outperformance.
- The paper doesn't provide sufficient details on how the FFDM and FIDM are interconnected.  Also more details on the action adapter at the output of the FIDM needs to be provided.
- Overall, the key insight in the paper is to break up a typical VLA network into two parts (FFDM, FIDM) with each of these being pretrained separately and then finetuned together.  This is neither a theoretical contribution nor much of a breakthrough in architecture.  In this sense, the novelty of the paper is limited.  Granted, they do claim a (slight) improvement over SOTA.

**Questions:**

- Unclear why the inverse dynamics only takes o_t and o_{t+n} and ignores all frames inbetween. Is the produced action then only at one time instant?
- It's unclear what outputs of the FFDM serve as inputs to the FIDM.  Are the o_t and o_{t+N} outputs from FFDM that serve as input to FIDM (particularly, o_t is an input to FFDM as well right?).  Fgiure 2(a, b) should be updated to showcase this connection.
- Authors should provide some detail on why increasing avg. length in CALVIN ABC-D benchmark is good (is this indicative of long-horizon tasks?).
- The authors only consider 3D action prediction.  I presume this is 3D end-effector position prediction (rather than joint angle prediction).  The authors should mention why this is the chos3en action to predict.
- Fig. 2(c) shows an action adapter - this is missing in 2(b) - authors should clearly indicate in 2(b) what the action adapter is.

---

> ### Author Response · Authors · 2025-11-21
> **Official Comment by Authors**
>
> We thank the reviewer for the insightful comments and constructive suggestions. Below we address each concern in detail. We hope our clarifications and new experiments resolve your concerns, and we would be very grateful if you could consider raising your score after reading this rebuttal.
>
> ### [**_W1_: The justification about significant outperformance.**]
>
> * **The 4.2% gain(4.33 → 4.51) on CALVIN ABC-D  is substantial, given that CALVIN evaluates the average success of five consecutive long-horizon tasks.** As shown in Table 1 in the manuscript, each stage improves by around 4%, and such consistent per-stage improvement is difficult to achieve due to error accumulation in long-horizon manipulation. Prior works typically show only marginal progress here, so a 4.2% gain represents a meaningful performance jump.
>
> * **The improvements stem from genuine methodological innovations rather than incremental tuning.** Our pretrained FFDM and FIDM introduce stronger dynamics priors and interactions, enabling more reliable long-horizon reasoning. As shown in Tables 4–7, each module contributes substantial gains (e.g., 3.28 → 4.51 in Table 4), confirming their causal impact.
>
> * **The significance of our design is further validated by consistent improvements across other benchmarks,** including SimplerEnv and real-world robotic evaluations, and by a 6.5% gain in our single-view setting.
> Together, these results demonstrate that the 4.2% improvement is both meaningful on a challenging benchmark and well-justified by the core innovations of our method.
>
> ### [**_W2.1_: The details on how the FFDM and FIDM are interconnected.**]
>
>  A lightweight MLP maps future latent embeddings from FFDM into the input space of the inverse dynamics model.
>
> We would rewrite it to emphasize the detail in Sec. 3.4 of the revision.
>
> ### [**_W2.2_: The details on action adapter.**]
>
> We build a 30M DiT-B based action adapter, and use the latent actions from the IDM as the condition of the diffusion-based to produce the corresponding executable action sequence.
>
> We would rewrite it to emphasize the detail in Sec.3.4 of the revision.
>
>
> ### [**_W3_: The novelty of this framework.** ]
>
> While previous works also decompose VLA models into two modules, our contributions go beyond this architectural split in two key aspects:
>
> * **Leveraging large-scale human videos for dynamics learning**:
> Unlike prior methods that rely heavily on scarce labeled robot data, our design enables effective use of unlabeled human videos by building a forward dynamics model capable of extracting rich dynamic priors from large-scale human motion. As shown in Fig. 4 and Table 5, this yields substantial performance gains.
>
> * **Revisiting the overlooked role of inverse dynamics**:
> We introduce a powerful inverse dynamics model and a self-supervised training framework, showing that accurate action recovery is as critical as accurate future prediction. Our analysis (Question 8 in the revision) demonstrates that incorrect action inversion accounts for 38% of failures on CALVIN, revealing a key limitation in prior approaches and highlighting the significance of our method.
>
> ### [**_Q1_: The confusion about the input of the inverse dynamics model.** ]
>
> * During pretraining, we train the FIDM with two frames(o_t, o_{t+n})  as input.
> * In the finetuning stage, we do not discard any frames. Instead, we use all the predicted frames {o_t, o_{t+1}, o_{t+2}, ..., o_{t+m}} and leverage the FIDM and action adapter to compute the executable actions between each consecutive frame.
>
> ### [**_Q2_: What outputs of the FFDM serve as inputs to the FIDM.** ]
>
> * During the pretraining stage, the input to the FIDM is the feature embedding extracted from images using DINO, rather than the output of the FFDM. Therefore, we did not include this connection in Fig. 2(a) and (b).
>
> * In the finetuning phase, the outputs of the FFDM are the video latent embeddings, which include the current frame and a sequence of future frames (e.g., \( o_t, o_{t+1},..., o_{t+n} \)). Based on these embeddings, FIDM predicts the corresponding action sequences.

---

> ### Author Response · Authors · 2025-11-21
> **Official Comment by Authors(Q3-Q5)**
>
> ### [**_Q3_: Some detail on why increasing avg. length in the CALVIN ABC-D benchmark is good.**]
>
> We have already discussed the difficulty and importance of increasing the average task length in the CALVIN ABC-D benchmark in Sec. 4.2. The longer task lengths are indicative of long-horizon tasks, which are more challenging and require better planning, decision-making, and generalization. These tasks are critical for evaluating the performance of models in real-world scenarios, where tasks often span over longer time horizons. We believe this addition strengthens our argument for the effectiveness of our approach in handling such tasks.
>
> ### [**_Q4_: The reason about predicting 3D action.**]
>
> * **Human Video Data**: Most of the human video data does not provide visibility into the shoulder joints, making it challenging to effectively predict joint angles. On the other hand, end-effector positions can still be reliably tracked and predicted even without joint angle information.
> * **Robot Data**: Different robots, such as the Google robot, WidowX, and Franka, have varying numbers of joints, making joint angle prediction less practical as a unified approach. By focusing on the 3D end-effector position, we ensure consistency across different robot types and simplify the prediction task.
> Thus, predicting 3D end-effector positions provides a more generalizable and practical solution for both human and robot video data.
>
>
> ### [**_Q5_: The confusion about Fig.2(b).**]
> We do not employ or pretrain the action adapter during the pretraining stage. Since we use a large amount of unlabeled human and robot video data, the focus during pretraining is on self-supervised learning to extract latent actions consistently. As a result, we do not require the use of an action adapter in this stage(Fig.2(b)).

---

> ### Author Response · Authors · 2025-11-27
>
> Dear Reviewer Hekh,
>
> I hope you are doing well.
>
> I am reaching out to kindly check in on the status of your review for our submission. We greatly appreciate the time and attention you have devoted to this process.
>
> If there is any further information or clarification we can provide to assist you, please do not hesitate to let us know.
>
> Thank you very much for your time and consideration.
>
> Best regards,
>
> The Authors of Submission 935

---

### Official Review · Reviewer_rVfL · 2025-11-01

**Soundness:** 2
**Presentation:** 2
**Contribution:** 2
**Rating:** 4
**Confidence:** 4

**Summary:**

This paper introduces DeFI, a framework where first a forward dynamics model and a latent inverse dynamics model is learned on various video data without actions, and then an action head is fine-tuned to map latent actions to ground truth actions in a specific embodiment. The forward dynamics model is a diffusion model in visual embedding space, the inverse dynamics model is a VQVAE, and the action head is a diffusion policy. A core claim is that pretraining forward and inverse dynamics models separately improves performance over coupled pretraining on actionless videos. Various experiments in sim and real environments show that DeFI generalizes better than previous state-of-the-art under the same downstream finetuning setup.

**Strengths:**

- The idea of learning forward and inverse dynamics for better generalization on video data is well-established.
- Disentangling forward/inverse dynamics learning and pretraining on large datasets is well-motivated.
- Various experiments in sim and real environments show that DeFI improves downstream policy performance over prior state of the art.
- Ablations validate the necessity of each design component in DeFI.

**Weaknesses:**

- The main concern that I have is end-to-end finetuning of forward/inverse dynamics models seem to undercut the claim that disentangled learning of forward/inverse dynamics model improves performance. It would be good to see a clarification on what the authors mean by coupled end-to-end finetuning, as well an ablation where only one of forward/inverse/action head is fine-tuned on downstream policy data. See question section below.
- No details on inference during robot experiments. The paper covers training and finetuning in detail in both the main text and the appendix. I might have missed it, but there doesn't seem to be much information on how exactly the forward dynamics/inverse dynamics/action head is then conditioned to complete tasks on the sim and real environments.

**Questions:**

- In section 3.3, what do the authors mean by finetuning the coupled FFDM and FIDM end-to-end in the title? If the core claim is disentangled forward/inverse dynamics learning, then this seems to undercut the claim; then Appendix A.2 says you freeze the forward dynamics model and only finetune the inverse dynamics model and action head. It would be nice to see an ablation / clarification.
- Could the authors clarify the inference process?
- Franka Play Dataset seems to have a wrong citation in Table 10.

---

> ### Author Response · Authors · 2025-11-21
>
> We sincerely thank you for your thoughtful comments on our paper. Below we address your concerns in detail, and we would be very grateful if you would consider adjusting your rating in light of these clarifications.
>
> ### [**_W1&Q1.1_:  The clarification on Coupled finetuning.**]
>
> * Pretraining (disentangled): FFDM and FIDM are trained separately, allowing each to use larger-scale data and simpler objectives, leading to stronger and more stable dynamics/action priors.
>
> * Coupled finetuning (clarification): “Coupled end-to-end” does not mean jointly updating both modules.
> FFDM is kept frozen because it has already been pretrained on large-scale data that includes the downstream dataset; further finetuning on the much smaller downstream splits would overwrite these dynamics priors and harm generalization. Therefore, only the FIDM and the action adapter are finetuned. The coupling serves two purposes:
>
>   * Align latent actions to the robot action space: The action adapter aligns the latent actions produced by the FIDM (from mixed human+robot pretraining) to the robot’s executable action space.
>
>   * Prevent mismatch between FFDM and FIDM: Because the frozen FFDM performs one-step denoising during inference while the FIDM was pretrained on DINO embeddings, the distribution of its inputs shifts. Therefore, the FIDM needs to be finetuned to align with the latent outputs of the FFDM.
>
> We have reorganized Sec. 3.3 to improve clarity and ensure that the reviewer’s concerns are fully addressed.
>
> ### [**_Q1.2_:  The ablation about the partial module training.**]
> We conduct the ablation study as shown in the following table(Tab.9 in the revision):
>
> | **Method**                           | **Task 1** | **Task 2** | **Task 3** | **Task 4** | **Task 5** | **Ave. Len.** |
> |--------------------------------------|------------|------------|------------|------------|------------|-------------------|
> | Adapter Only |  95.5 | 92.0 | 87.2 | 81.1 | 75.7 | 4.33 |
> | FDM+Adapter  | 95.6 | 92.4 | 88.1 | 82.5 | 76.2 | 4.35 |
> | **IDM+Adapter(Ours)**  | **97.9** | **94.2** | **90.7** | **87.0** | **81.2** | **4.51** |
> | All Train    | 96.8 | 93.1 | 88.4 | 83.2 | 78.0 | 4.40 |
>
> **Analysis:**
> Adapter Only already achieves strong performance despite having very few trainable parameters, indicating that the pretrained FFDM provides a strong and expressive latent space. FDM+Adapter yields only minor improvements, suggesting that forward prediction alone is insufficient for reliable action inference. Although “All Train” updates FFDM, IDM, and the action adapter jointly, its performance is lower than IDM+Adapter(Ours) because joint optimization introduces representation instability and gradient interference. When FFDM is finetuned, its latent outputs change throughout training, causing the input distribution of IDM to drift. As a result, the IDM must continually adapt to shifting representations, making it much harder to learn a stable and accurate action-recovery function. Additionally, the frozen forward dynamics model could provide better generalization for action reasoning.
>
>
> ### [**_W2&Q2_:  The details of inference.**]
>
> At inference time, the FFDM receives the current observation o_t and language instruction l, generates a sequence of predicted future video features \hat{z}_{t:t+H} through a single-step denoising process.
> These features capture the anticipated future scene evolution and establish a dynamics-aware context for downstream action reasoning.
> The MLP then projects these future embeddings into the input space of the FIDM, which combines them with the current latent state to infer the latent action sequence
> The diffusion-based action adapter then conditions on these latent actions and produces the final executable control commands.
>
>
> We have added the details of the inference stage in the revision.
>
> ### [**_Q3_: The wrong citation.**]
>
> We have corrected this wrong citation in the revision.

---

> ### Author Response · Authors · 2025-11-27
>
> Dear Reviewer rVfL,
>
> I hope this message finds you well.
>
> I am writing to kindly follow up on the status of your review for our submission. We truly appreciate the time and effort you have invested, and we understand your schedule may be very busy.
>
> If there is any additional information or clarification needed from our side to facilitate the process, please feel free to let us know — we would be glad to assist.
>
> Thank you very much for your time and consideration.
>
> Best regards,
>
> The Authors of Submission 935

---

### Official Review · Reviewer_2pYP · 2025-11-11

**Soundness:** 3
**Presentation:** 3
**Contribution:** 3
**Rating:** 6
**Confidence:** 4

**Summary:**

This paper introduces DeFI, a novel framework that decouples visual forward dynamics and inverse dynamics pretraining to better leverage large-scale action-free videos for robot learning. The key innovation lies in separately pretraining two components: (1) a Foundation Forward Dynamics Model (FFDM) via video generation on mixed human/robot videos, and (2) a Foundation Inverse Dynamics Model (FIDM) using self-supervised learning to extract latent actions from video transitions without requiring explicit action labels. These models are then coupled and fine-tuned end-to-end for downstream tasks. The approach achieves state-of-the-art results on CALVIN ABC-D (4.51 average task length), SimplerEnv-Fractal (51.2% success rate), and real-world experiments (81.3% success rate).

**Strengths:**

- Motivation: The paper clearly articulates the fundamental misalignment between 2D video forecasting and 3D action prediction in current VLA approaches, making a compelling case for the decoupled approach.

- Presentation: The writing is accessible, the motivation is well-articulated, and the figures effectively illustrate both the method and results. The paper flows logically from problem identification to solution.

- Experimental validation: The authors provide extensive experiments across multiple benchmarks (CALVIN, SimplerEnv, real-world Franka) with consistent improvements demonstrated across all settings.

- Ablations: Tables 4-7 systematically validate multiple design choice, from the importance of pretraining to architectural decisions, providing good insights into what drives performance.

- Method: The approach offers an interesting path toward leveraging action-free data for training, with a novel inference mechanism that combines forward and inverse dynamics in a principled way.

**Weaknesses:**

- Outdated baselines: The comparison baselines don't include the most recent state-of-the-art VLAs, which diminishes the impact of the results. The paper would be significantly strengthened by comparisons against more recent models like Gr00t or π0/π0.5.

- Frozen FFDM limitations: The authors acknowledge that the frozen FFDM causes performance issues on SimplerEnv due to sim-to-real gaps, which appears to be a fundamental limitation of the approach that isn't adequately addressed.

- Limited gains from human videos: Table 5 shows only modest improvements from incorporating human videos (+0.17 on average task length). Given the added complexity of the dual pretraining pipeline, it's unclear whether this marginal gain justifies the approach.

**Questions:**

- VQ-VAE discretization: Why specifically does VQ-VAE discretization help inverse dynamics learning? Have you experimented with other discretization methods (Gaussian mixture models, simple binning) or continuous latent actions? The paper would benefit from more analysis on why this particular bottleneck design is optimal.

- DINO-based world model: You briefly mention a DINO-based world model in Table 6. Could you elaborate on why this underperforms the pixel-based approach? Intuitively, predicting future DINO latents with regression loss seems appealing - it would align better with the FIDM input space and reduce inference time.

- Scaling behavior: How does performance scale with pretraining data size? Is there a point of diminishing returns? This is particularly important given the main motivation is to leverage large-scale human data.

- Single denoising step: While you show that one denoising step maintains task performance (Table 6), can you provide visual comparisons showing what motion information is preserved versus lost? This would help understand why this aggressive optimization works.

- Failure modes: What are the primary failure cases? Does the model struggle more with forward dynamics prediction or inverse dynamics inference? Qualitatively, where does this approach excel compared to standard VLAs, and where do VLAs maintain advantages (perhaps in reactive behaviors given their faster inference)?

- Domain adaptation: Have you explored partial fine-tuning or adapter layers for FFDM that could address domain shift while preserving the benefits of pretraining? This question is relvant for the ` Frozen FFDM limitations` raised aboce

---

> ### Author Response · Authors · 2025-11-21
> **Author Response (Weakness1-Weakness3)**
>
> We thank the reviewer for the insightful comments and constructive suggestions. Below we address each concern in detail. We hope our clarifications and new experiments resolve your concerns, and we would be very grateful if you could consider raising your score after reading this rebuttal.
>
> ### [**_W1:_ Lack of comparison with the most recent state-of-the-art VLAs.]**
>
> We have already compared against π0 in Tab.1 of the manuscript. Following the reviewer’s suggestion, we additionally include π0.5 and GR00T N1 results on CALVIN, shown below (this will be added as Table 1 in the revision):
>
> | Addition Type    | 1    | 2    | 3    | 4    | 5    | Len. |
> |------------------|------|------|------|------|------|------|
> |π0 |93.8 | 85.0 | 76.7| 68.1 | 59.9| 3.84|
> |π0.5     | 94.8 | 87.4 | 78.2 | 71.7 | 64.3 | 3.97 |
> | GR00T N1    | 94.2 |86.1| 79.6| 73.9 |66.8 | 4.01 |
> | OpenVLA-OFT | 95.7 | 88.2 | 82.4 | 74.5 | 70.7 | 4.12 |
> |**DeFI(Ours)**|**97.9** | **94.2** | **90.7** | **87.0**| **81.2** |**4.51**|
>
> **Analysis:**
> Our method significantly outperforms π0.5 and GR00T N1, both of which directly project RGB observations into actions. This comparison highlights a key advantage of DeFI: **pretraining on large-scale mixed data provides strong prior knowledge, and the forward dynamics model predicts future states to supply richer guidance for action inference**, leading to substantially better performance, which has been verified by the concurrent works like CoT-VLA[1], DreamVLA[2],  VPP[3].
>
> >[1].Zhao, Qingqing, et al. "Cot-vla: Visual chain-of-thought reasoning for vision-language-action models." Proceedings of the Computer Vision and Pattern Recognition Conference. 2025.
>
> >[2].Zhang, Wenyao, et al. "DreamVLA: A Vision-Language-Action Model Dreamed with Comprehensive World Knowledge." The Thirty-ninth Annual Conference on Neural Information Processing Systems.
>
> >[3].Hu, Yucheng, et al. "Video Prediction Policy: A Generalist Robot Policy with Predictive Visual Representations." Forty-second International Conference on Machine Learning.
>
> ### [**_W2_: The confusion about the frozen FFDM.]**
>
> Freezing FFDM is an intentional and principled design choice, rather than a limitation. The reasons are as follows:
>
> * **1. The “sim-to-real gap” originates from SimplerEnv’s Real2Sim pipeline.**
> SimplerEnv is built by converting real robot demonstrations into a simulator; the rendered frames exhibit a significant visual domain gap from real images. Since FFDM predicts future frames from the simulator’s initial observation, this mismatch leads to a discrepancy between FFDM-predicted future frames and actual simulator observations. This issue is specific to SimplerEnv. In settings where training and testing share the same modality (real-world or CALVIN), our method performs consistently well.
>
> * **2. Freezing FFDM is essential for generalization.**
> Because it was pretrained on large-scale data that already covers the downstream domain, further finetuning on the much smaller downstream split would erode these dynamics priors and hurt generalization. Even for unseen objects, fronzen FFDM can still produce plausible future predictions, enabling FIDM to infer correct actions, which is consistent with [1][2]. Finetuning FFDM on limited downstream data would overfit to the specific environment, reduce generalization to new objects/scenes, and undermine the goal of building a foundation dynamics model.
>
> >[1].Feng, Yao, et al. "Vidar: Embodied Video Diffusion Model for Generalist Manipulation." arXiv preprint arXiv:2507.12898 (2025).
>
> >[2].Hu, Yucheng, et al. "Video Prediction Policy: A Generalist Robot Policy with Predictive Visual Representations." Forty-second International Conference on Machine Learning.
>
>
>
> ### [**_W3_: Limited gains from human videos.]**
>
> The +0.17 in Table 5 reflects only the case where FIDM is trained without human videos, while FFDM still uses human-video pretraining. This does not capture the full benefit. When both FFDM and FIDM remove human-video pretraining, performance drops(4.51->3.97) as shown in the following table(Tab.5 in the revision), showing a clear and significant contribution from human videos.
>
> | Addition Type    | 1    | 2    | 3    | 4    | 5    | Len. |
> |------------------|------|------|------|------|------|------|
> | All w/o h.v.     | 93.6 | 91.2 | 88.0 | 82.4 | 79.2 | 3.92 |
> | FFDM w/o h.v.    | 96.0 | 91.2 | 85.6 | 77.6 | 68.8 | 4.19 |
> | FIDM w/o h.v.    | 93.6 | 91.2 | 88.0 | 82.4 | 79.2 | 4.34 |
> | **All w/ h.v. (Ours)**  | **97.9** | **94.2** | **90.7** | **87.0** | **81.2** | **4.51** |

---

> ### Author Response · Authors · 2025-11-21
> **Author Response (Question1-Question4)**
>
> ### [**_Q1_: Different VQ-VAE discretization.**]
>
> VQ-VAE is used not merely as a discretization tool but as an information-bottleneck mechanism that stabilizes inverse dynamics learning. The quantization step forces the model to compress both the current and future frame embeddings, which prevents future-state leakage into the decoder.  We evaluate other discretization strategies(Tab.8 in the revision):
>
> | Addition Type              | 1    | 2    | 3    | 4    | 5    | Len. |
> |---------------------------|------|------|------|------|------|------|
> | Gaussian Mixture          | 94.1 | 90.3 | 84.7 | 78.5 | 72.9 | 4.12 |
> | Simple Binning            | 93.2 | 89.1 | 82.3 | 75.4 | 69.8 | 3.98 |
> | Continuous Latent Action  | 94.5 | 91.0 | 86.2 | 80.1 | 74.7 | 4.20 |
> | **Discrete (Ours)**       | **97.9** | **94.2** | **90.7** | **87.0** | **81.2** | **4.51** |
>
> **Analysis:**
> VQ-VAE provides the best trade-off between stability, action consistency, and downstream task performance, which is why we adopt it as the default design. These alternatives either lead to training instability (due to insufficient information compression) or produce less structured latent actions, resulting in inferior downstream performance.
>
> ### [**_Q2_: Why not use DINO-based world model.**]
>
> The DINO-based world model underperforms because **it can not benefit from powerful pretrained video generative models, whose training data is far larger and more diverse than our own**, providing strong dynamic priors that DINO-regression models cannot access.
> While a DINO-based model can indeed use human video data, it lacks these pretrained generative priors and thus gains much less from large-scale human videos. We view DINO and other latent feature–based generation as a promising direction. We plan to explore stronger DINO-based prediction models (e.g., DINO-WM [1]) as well as other latent embedding prediction models (e.g., V-JEPA [2], V-JEPA 2 [3]) to better understand the upper bound of DINO and related latent features in the future. We have added this discussion in the revision.
>
> > [1]. Zhou, Gaoyue, et al. "Dino-wm: World models on pre-trained visual features enable zero-shot planning." arXiv preprint arXiv:2411.04983 (2024).
>
> >[2]. Bardes, Adrien, et al. "V-jepa: Latent video prediction for visual representation learning." (2023).
>
> >[3]. Assran, Mido, et al. "V-jepa 2: Self-supervised video models enable understanding, prediction and planning." arXiv preprint arXiv:2506.09985 (2025).
>
>
> ### [**_Q3_: Scaling behavior.**]
>
> Our primary goal in this work is to demonstrate the feasibility and effectiveness of leveraging human videos for dynamics pretraining. Although the scaling-law study is beyond the scope of this paper, we include additional experiments at the reviewer’s request as shown in the following table (Tab.7 and Fig.8 in the revision):
>
>
> | **Scale** | **Task 1** | **Task 2** | **Task 3** | **Task 4** | **Task 5** | **Ave.Len.** |
> |------------|------------|------------|------------|------------|------------|-------------------|
> | 0.0   | 92.4   | 85.6   | 78.0   | 70.2   | 63.1   | 3.92 |
> | 0.2   | 94.6   | 88.2   | 83.3   | 77.5   | 71.9   | 4.16 |
> | 0.4   | 96.4   | 91.0   | 86.0   | 80.7   | 75.4   | 4.30 |
> | 0.6   | 96.1   | 92.4   | 87.4   | 82.6   | 77.3   | 4.36 |
> | 0.8   | 97.5   | 93.3   | 89.2   | 84.1   | 78.9   | 4.43 |
> | **1.0**    | **97.9**   | **94.2**   | **90.7**   | **87.0**   | **81.2**   | **4.51**          |
>
> **Analysis:**
> Performance continues to increase as the amount of human video rises. While the marginal gains become smaller at larger scales, we do not observe saturation, indicating that the current scale is far from the upper limit. We plan to extend our pretraining to substantially larger human-video corpora to further analyze scaling trends, and we believe these results will provide useful insights for future research on data scaling in robot learning.
>
> ### [**_Q4_: Visualization of single denoising step**]
>
> As shown in Fig.9 in the revision, although a single denoising step may blur background regions, we observe that **the motion-critical information—such as the object trajectory and the robot end-effector path—is still well preserved**, which explains why task performance remains stable. In addition, **the finetuning of the inverse dynamics model and the adapter further mitigates potential information loss**, enabling the model to operate reliably even under this aggressive optimization.

---

> ### Author Response · Authors · 2025-11-21
> **Author Response (Question5-Question6)**
>
> ### [**_Q5.1_: Failure case analysis.]**
>
>  We inspected 200 failure cases on CALVIN (failures defined as not completing the task within 280 steps). The errors fall into two categories:
> * **Forward Dynamics Model Failures (62%)**: As shown in Fig.10 in the revision, FFDM often struggles in contact-rich interactions and multi-view consistency, occasionally producing hallucinated predictions. Once the interaction phase ends, its predictions become accurate again.
> * **Inverse Dynamics Model Failures (38%):** As shown in Fig.11 in the revision, even with good future predictions, the IDM may still output incorrect actions (e.g., misplacing objects or causing collisions).
> This highlights our motivation: accurate inverse dynamics is as essential as accurate forward prediction for reliable control.
>
>
> ### [**_Q5.2_: comparison with standard VLA.]**
>
> **Advantage**:
> * Explicit reasoning via future prediction: video-based world models allow the agent to predict future states before acting, providing explicit visual rollouts rather than relying solely on VLM text-token prediction. This yields more structured and accurate decision-making.
>
> * Simplified action inference & better pretraining signal:  by decoupling the problem into (i) future prediction and (ii) inverse action inference, the learning task becomes easier to optimize. Moreover, FFDM can leverage large-scale action-free human video, giving stronger dynamics priors than action-conditioned VLA models.
>
> **Disadvantage**:
>
> * Sometimes the forward dynamics model produces inaccurate future predictions.
>
>
>
> ### [**_Q6_: The exploration for partial tuning of FFDM.]**
> In fact, LoRA improves the small in-domain performance gains on CALVIN, confirming that lightweight adaptation can help mitigate domain shift. However, it also degrades the generalization, suggesting that even minor updates can overwrite the broad dynamics priors learned during large-scale pretraining. This supports our choice to keep FFDM frozen, while considering adapter-based approaches as promising future work.

---

> ### Author Response · Authors · 2025-11-27
>
> Dear Reviewer 2pYP,
>
> I hope this message finds you well.
>
> This is a gentle follow-up regarding your review of our submission. We sincerely appreciate the time and effort you have dedicated to the process.
>
> Please let us know if you require any additional information or clarifications from our side.
>
> Thank you for your consideration.
>
> Best regards,
>
> The Authors of Submission 935

---

### Author Response · Authors · 2025-11-30
**General response**

We sincerely thank the reviewers and AC for their time, constructive feedback, and thoughtful comments. Our paper demonstrates that a carefully decoupled forward/inverse dynamics framework enables effective use of large-scale action-free human videos, then stably couples them to enable end‑to‑end finetuning, yielding strong manipulation performance on different simulators and real-world experiments.

We are encouraged that all reviewers recognized DeFI as an effective method for fully leveraging large-scale action-free data. We also appreciate the recognition of clear motivation and conceptual novelty (Reviewers 2pYP, rVfL, zL2E), as well as the acknowledgment of our comprehensive experiments and ablation studies (Reviewers 2pYP, rVfL, zL2E).
To address remaining concerns, we made the following concrete updates in the revision:

**Stronger baselines & more experiments.** Added π0.5, OpenVLA‑OFT, GR00T‑N1 under the same splits/protocol. DeFI remains SOTA on CALVIN ABC‑D (Avg. Len. 4.51), ahead of OpenVLA‑OFT (4.12). (Reviewer 2pYP, zL2E)

**Clarified pipeline & rationale.** Reorganized Sec. 3.3&3.4 with concrete I/O, the alignment MLP, the diffusion action adapter, and why FFDM is frozen (to preserve large‑scale dynamics priors—including CALVIN in pretraining—and avoid representation drift that hurts IDM). (Reviewer (Hekh,rvfL))

**Targeted ablations answering the questions raised.**

* Human‑video scaling: monotonic gains without saturation at our scale. (Reviewer 2pYP)

* Discretization for IDM: VQ‑VAE bottleneck > Gaussian Mixture / Simple Binning / Continuous latents. (Reviewer 2pYP)

* Partial‑module tuning: IDM+Adapter > All‑Train; freezing FFDM stabilizes action inversion. (Reviewer rVfL)



Notes to AC. We also revised the main text accordingly(blue text), and these updates directly resolve the fairness and clarity concerns.
As noted in the discussion, the reviewer zL2E who raised the fairness concern has confirmed that they will revise their score upward once the requested additional comparisons are included. We have now completed these experiments (e.g., OpenVLA‑OFT) and incorporated the results in the revision.

---

### Meta-Review · Area_Chair_dWof · 2026-01-04

**Summary:**

My recommendation for this paper is Accept. This paper was at the very borderline. As Reviewer Hekh pointed out, the paper is not proposing a completely new framework or advancing the field with surprising results. However, the paper's technical contributions are valid, motivation is clear, the paper has no strong weaknesses, and it is a combination of solid ideas and good execution pipelines. I recommend the authors to further sharpen the claims on the contributions and novelties of the proposed framework and improve the draft for the camera-ready version.

**Reviewer Concerns:**

Concerns addressed by the rebuttal
- Missing baselines (pi0.5, Gr00t, OpenVLA-oft)
- Clarification on whether the paper is fine-tuning the baselines as well
- Clarification on gains from human videos
- Clarification on the setup (not fine-tuning forward dynamics model)
- Missing details on robot inference
- Difference to prior work

Outstanding concerns
- Limited novelty: While the authors highlighted two main differences (`Leveraging large-scale human videos for dynamics learning` and `Revisiting the overlooked role of inverse dynamics`), I find this claim quite weak; as learning from large-scale videos and using inverse dynamics is being widely studied in the community

**Reviewer Scores:**

- Reviewer 2pYP (score 6): I expect this reviewer to maintain their score
- Reivewer rVfL (score 4): I expect this reviewer to increase the score to 6
- Reviewer Hekh (score 6): I expect this reviewer to maintain their score
- Reviewer zL2E (score 4): I expect this reviewer to increase their score to 6

---

### Decision · Program_Chairs · 2026-01-26

Accept (Poster)